# Global critical soil moisture thresholds of plant water stress

Zheng Fu [1,2] ✉, Philippe Ciais [2], Jean-Pierre Wigneron [3], Pierre Gentine [4], Andrew F. Feldman [5,6], David Makowski [7], Nicolas Viovy [2], Armen R. Kemanian [8], Daniel S. Goll [2], Paul C. Stoy[9], Iain Colin Prentice [10,11], Dan Yakir [12], Liyang Liu [2], Hongliang Ma[13], Xiaojun Li[3], Yuanyuan Huang [1], Kailiang Yu [2], Peng Zhu[14], Xing Li [15], Zaichun Zhu [16], Jinghui Lian [2] & William K. Smith [17]

During extensive periods without rain, known as dry-downs, decreasing soil moisture (SM) induces plant water stress at the point when it limits evapotranspiration, defining a critical SM threshold ($\theta_{crit}$). Better quantification of $\theta_{crit}$ is needed for improving future projections of climate and water resources, food production, and ecosystem vulnerability. Here, we combine systematic satellite observations of the diurnal amplitude of land surface temperature (dLST) and SM during dry-downs, corroborated by in-situ data from flux towers, to generate the observation-based global map of $\theta_{crit}$. We find an average global $\theta_{crit}$ of 0.19 m³/m³, varying from 0.12 m³/m³ in arid ecosystems to 0.26 m³/m³ in humid ecosystems. $\theta_{crit}$ simulated by Earth System Models is overestimated in dry areas and underestimated in wet areas. The global observed pattern of $\theta_{crit}$ reflects plant adaptation to soil available water and atmospheric demand. Using explainable machine learning, we show that aridity index, leaf area and soil texture are the most influential drivers. Moreover, we show that the annual fraction of days with water stress, when SM stays below $\theta_{crit}$, has increased in the past four decades. Our results have important implications for understanding the inception of water stress in models and identifying SM tipping points.

The critical soil moisture threshold ($\theta_{crit}$) of plant water stress is defined as the soil moisture (SM) level at which evapotranspiration becomes SM limited in that environment[1]. Below this threshold, a marginal reduction of SM reduces evapotranspiration and increases sensible heat emissions and surface temperature[2], making the air above the canopy warmer and drier, which in turn further reduces evapotranspiration and plant carbon dioxide uptake[3–5]. The control of energy partitioning regimes across $\theta_{crit}$ determines local climate through land-atmosphere coupling and can amplify warming during droughts[6,7]. A better knowledge of $\theta_{crit}$ is thus important for land-atmosphere interactions[5], for climate studies[8–10] and for understanding the vulnerability of ecosystems and crop yields to drought[9].

The relationship between SM and the evaporative fraction (EF), defined as the ratio of evapotranspiration to net radiation, shows two distinct regimes[2,5,11,12] (see Fig. 1a). When SM is higher than $\theta_{crit}$, the system is non-water limited (energy limited) and SM does not impact evapotranspiration[5]. In contrast, when SM is lower than $\theta_{crit}$, the capacity of plants to extract soil water by roots and xylem transport becomes progressively reduced. The system becomes SM limited, and evapotranspiration decreases with decreasing SM until leaves fully close their stomata, direct evaporation at the soil surface ceases, or roots are no longer able to take up soil water (the wilting point)[10]. The overall EF–SM relationship (increasing below $\theta_{crit}$ and then plateauing) is conceptually well established, but a spatially explicit understanding

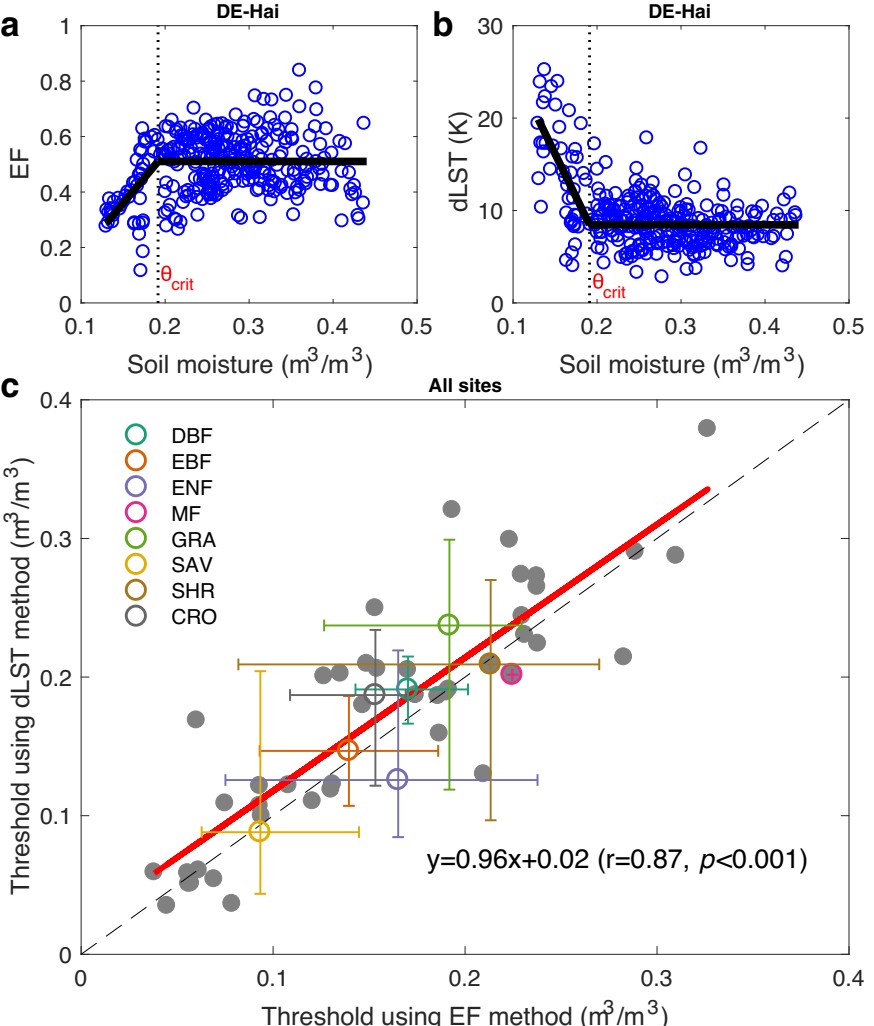

**Fig. 1 | Quantifying the critical soil moisture (SM) threshold during soil dry-downs using the evaporative fraction (EF)−SM method and the land surface temperature diurnal amplitude (dLST)−SM method.** An example of estimating SM thresholds ($\theta_{crit}$) from the EF−SM method (**a**) and the dLST−SM method (**b**) using all dry-downs at Hainich beech forest site (DE-Hai, Supplementary Table 1). **c** Comparison between the SM thresholds estimated from the dLST−SM method and EF−SM method across all sites. The median and the 25th, 75th percentiles are shown for each biome. The dashed line is the 1:1 line while the red line is fitted line using least squares regression.

with accurate global maps of $\theta_{crit}$ are lacking, due to a lack of global high-frequency observations of EF[11,13–15]. Therefore, the factors that control the global variations in $\theta_{crit}$ are poorly known. Earth system modelers have adopted simple parametric representations of EF-SM relationship and $\theta_{crit}$ to describe soil water stress and land-atmosphere feedbacks[5], leading to model biases which hinder our ability to predict drought and its ecosystem impacts[8,16–18]. Some model-based analyses have used the concept of critical soil water potential[1,19], but current land surface models are using soil moisture rather than soil water potential, and global observation-based analyses of critical thresholds are still missing.

Satellite observations of surface SM with frequent revisit and global coverage based on microwave sensors in the L-band with stronger penetration capacity[20] can be combined with land surface temperature (LST) to assess the relationships between SM and the surface energy partitioning[21,22] (Methods). Instead of global satellite evapotranspiration products based on models with uncertain parametrizations[23,24], we used here the diurnal evolution of LST as a direct observable signature of shifts in surface energy partitioning regimes[25,26]. Specifically, the land-surface temperature diurnal amplitude (dLST) starts to increase below $\theta_{crit}$ when ecosystems plunge into

the water-limited regime[26–29]. An increased dLST, for a given amount of net radiation, is directly linked to a decrease in EF and thus increased SM stress[3]. dLST is positively associated with sensible heating but negatively associated with EF and SM[27–29]. Evaporative regimes have been characterized with observed dLST−SM relationships across Africa, but not yet globally[14], leaving a gap in our understanding of $\theta_{crit}$ across the globe.

To quantify the global spatial distribution of $\theta_{crit}$, we selected extensive periods without rainfall known as SM dry-downs[14,30,31] when the transition from energy to water limitation is likely to happen. The validity of the dLST−SM approach to determine $\theta_{crit}$ was demonstrated by comparing its results to the classical EF−SM method[2,5,9,10] at sites of the global network of flux tower measurements (Methods). Three global satellite SM datasets (SMAP-IB, SCA-V and SMOS-IC) and two LST datasets (Copernicus and MODIS) covering the period from April 2015 to December 2020 were then used to produce a global map of $\theta_{crit}$. Uncertainties were estimated based on an ensemble of 18 members from different pairings of SM and LST datasets, including the uncertainty on $\theta_{crit}$ from the dLST−SM relationship (Methods). Additionally, explainable machine learning models (random forest) were applied to gain insights on the climatic, biotic and edaphic factors

controlling the spatial variations of $\theta_{crit}$. Based on our global map of $\theta_{crit}$, we further calculated the fraction of days in a year when SM is below $\theta_{crit}$ using time series of SM from satellite data and the ERA5-Land reanalysis[32], to investigate the long-term trends of plant exposure to water stress over the last 40 years. Finally, we evaluated how land surface models of Earth System Models participating in the Coupled Model Inter-comparison Project Phase 6 (CMIP6) simulate the patterns of $\theta_{crit}$, and discussed their biases compared to our observation-based maps.

## Results and discussion

### Consistency of $\theta_{crit}$ derived from the EF−SM and dLST−SM methods

Using soil dry-downs observed at 44 flux tower sites, we calculated daily dLST using in-situ daily maximum and minimum outgoing long-wave radiation (Methods) and compared $\theta_{crit}$ defined as the break-point when dLST increases with decreasing SM, with the value calculated from the flux data as a tipping point of the EF−SM relationships, as previously done in refs. 2,5,9,10. An example is shown for the Hainich beech forest site in Germany (Supplementary Table 1) where the EF−SM relationship during dry-downs defines a $\theta_{crit}$ of $0.192 \pm 0.009$ m³/m³ (± standard error) and the dLST−SM relationship gives a very similar estimate of $0.191 \pm 0.005$ m³/m³ (Fig. 1a & b). Across all the sites spanning a large range of aridity and plant functional types, the two approaches show consistent results ($r = 0.87$, Fig. 1c), in line with previous theoretical and observational studies[8,14,25].

### Global distribution of $\theta_{crit}$

Global high frequency LST and SM observations from multiple satellites during drydowns are then used to quantify the spatial distribution of $\theta_{crit}$ (Methods). To calculate daily dLST, we used the MODIS Terra and Aqua satellites and the Copernicus dataset based on a constellation of geostationary satellites. Even though MODIS only passes over the earth four times a day while the geostationary satellites data (Copernicus) have 24 observations per day, allowing us to define more accurately the diurnal amplitude of LST, both observations match well with each other (Supplementary Figs. 1–2). Over Siberia and India where no geostationary data are available, we used only MODIS. For quasi-daily SM, we used satellite all-weather data from SMAP-IB, SCA-V and SMOS-IC, which show a similar pattern of the number of dry-downs per year over each point of the Earth (Supplementary Figs. 3–4). Here, the SM drydowns were defined as periods with at least five (SMAP-IB and SCA-V) or four (SMOS-IC) consecutive overpass masurements over intervals longer than 10 days during which SM is persistently decreasing (Methods). The areas with the the largest number of dry-downs are in central America, Argentina, central Europe, eastern Europe and eastern Australia (Supplementary Fig. 3). On the other hand, only few dry-downs could be used to infer $\theta_{crit}$ in wet regions such as the Amazon, central Africa and southern China. Grid points with no clear dry-downs to calculate $\theta_{crit}$ were masked.

The global maps of $\theta_{crit}$ obtained with the three SM and the two dLST satellite datasets show consistent patterns (Fig. 2a–f). Different maps from paired SM and dLST observations with three $\theta_{crit}$ estimates (mean, and plus or minus one standard error, see Methods) provide an ensemble of 18 members. The median and standard error across all members of the ensemble shown in Fig. 2g represent our best estimate of the global distribution $\theta_{crit}$ and its uncertainty. The relative uncertainty of $\theta_{crit}$, defined as the ratio of standard error to the median value of the 18 ensemble members, is less than 10% over most areas (Fig. 2h). Moreover, despite a mismatch in spatial scales, the value of $\theta_{crit}$ from satellites at a global resolution of 25 by 25 km, is significantly correlated to the local estimate calculated at point-scale flux tower measurements (Supplementary Fig. 5). The median value of $\theta_{crit}$ over the global vegetated areas is 0.19 m³/m³. Even though satellites only probe surface SM, whereas plants may be sensitive to stress from surface and rootzone moisture deficits, surface SM has been shown to be equally skillful for identifying evapotranspiration regime changes as deeper soil layers or rootzone SM measurements[33,34]. We further use SM data with different soil layers from ERA5-Land (Methods) and compare the $\theta_{crit}$ values derived from ERA5-Land SM layer 1 (0−7 cm depth), layer 2 (7−28 cm) and layer 3 (28−100 cm). We found that surface $\theta_{crit}$ is highly correlated with $\theta_{crit}$ derived from deep soil layers (Supplementary Fig. 6), showing that $\theta_{crit}$ obtained from surface SM can provide information deeper into the subsurface, consistent with the results of flux tower observations reported by both Dong, Akbar[33] and Fu, Ciais[35].

The lowest $\theta_{crit}$ values were observed in dryland ecosystems over the western United States, western Argentina, eastern Brazil, South Africa, northwestern China and Australia (Fig. 2g). In those dryland regions, plant hydraulic features adapted to conditions when evaporative demand often exceed soil water supply, are likely to minimize $\theta_{crit}$ through mechanisms of sustained SM extraction by roots and transport by xylem, even at low soil water potentials[36]. Conversely, the highest $\theta_{crit}$ values were found in humid ecosystems such as Indonesia, south-eastern China, south-eastern United States, and Uruguay (Fig. 2g). Differences of $\theta_{crit}$ between biomes were found to be significant, with increasing $\theta_{crit}$ from dry shrublands, grasslands, and savannas towards temperate, boreal and tropical forests (Supplementary Fig 7a). Similar patterns were found across climate types, with increasing $\theta_{crit}$ from hyper-arid, arid, and semi-arid ecosystems towards humid ecosystems (Supplementary Fig. 8).

We performed a more detailed analysis of the $\theta_{crit}$ differences between cropland types, based on the expectation that $\theta_{crit}$ should be affected by the choice of cultivars and by management practices such as irrigation. We found that $\theta_{crit}$ varied among different crop species (Supplementary Fig. 7b), with rice (mostly irrigated) having significantly higher values (0.28 m³/m³) than maize, wheat and potato (0.20 m³/m³, $p < 0.05$). Moreover, $\theta_{crit}$ tended to increase with increasing irrigation (Supplementary Fig. 7c). We also tested the hypothesis that the areas of recent cropland expansion over drier marginal lands should be associated with a decrease of $\theta_{crit}$ (Methods). Most new cropland expansion occurred over drier areas, such as in southern Sahel, central Highlands and Zambia over Africa, and the Cerrado and Chaco plains in South America[37]. Over the 'new' cropland areas that were cultivated after 2003 according to the high resolution map of ref. 37, we verified that $\theta_{crit}$ was lower on average than on established cropland areas (Supplementary Fig. 7d). More insights on regional patterns of $\theta_{crit}$ and its impacts for yields could be gained based on regional management and cultivars information, which is beyond the scope of this study.

The spatial distribution of $\theta_{crit}$ in this study aligns with previous findings in ecological theory regarding plant stress across various environments[1,14,19,33,38–40]. Land surface models often have a lower $\theta_{crit}$ model parameter in arid biomes[38,40,41]. The map of ecosystem-scale isohydricity from remotely sensed observations showed that the anisohydric behavior is more common in arid ecosystems[39]. By quantifying the soil water potential threshold, Bassiouni, Good[1] showed that water uptake strategies in arid locations are generally more drought resistant. Note that soil water potential is rarely measured in situ, and land surface models are using soil moisture rather than soil water potential. Different vegetation water stress in arid and humid ecosystems have also been recognized in many other studies, based on the ecosystem limitation index[38], the Land Surface Water Index[42,43], and SM anomalies[40]. However, these indicators are not direct measures of water stress. The $\theta_{crit}$ values quantified in our study reflect the long-term adaptation of ecosystems to aridity regimes. $\theta_{crit}$ is simple to define and is a direct measure of water stress, but $\theta_{crit}$ remains not observed and our study allows to compare it across biomes. $\theta_{crit}$ can also be used to quantify the time spent below $\theta_{crit}$ and understand how recent climate trends have affected the exposure of ecosystems to water stress.

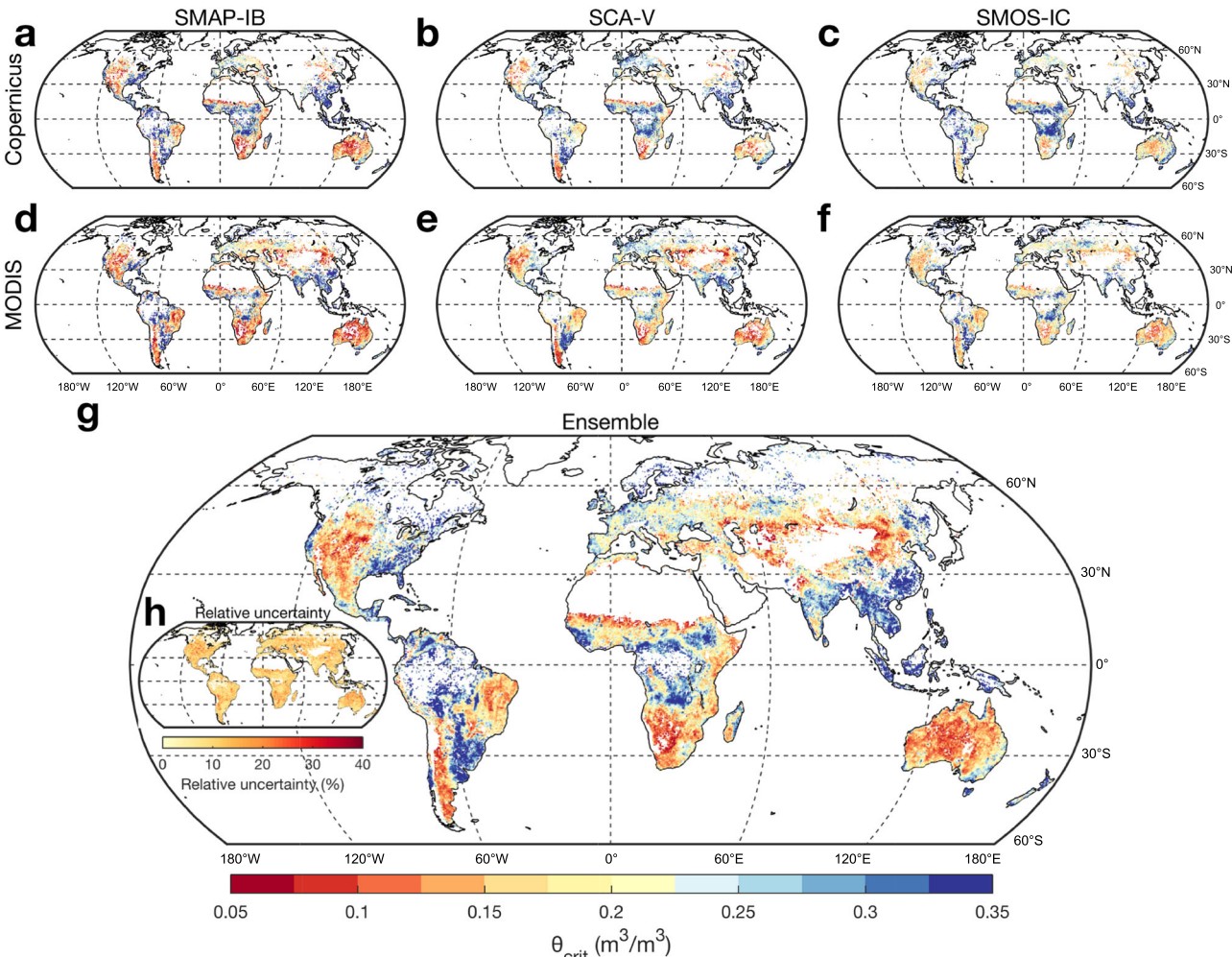

**Fig. 2 | The global distribution of critical soil moisture threshold ($\theta_{crit}$).** The global distribution of estimated $\theta_{crit}$ using Copernicus land surface temperature diurnal amplitude (dLST) and soil moisture (SM) from SMAP-IB (**a**), SCA-V (**b**) or SMOS-IC (**c**). The global distribution of estimated $\theta_{crit}$ using MODIS dLST and SM from SMAP-IB (**d**), SCA-V (**e**) or SMOS-IC (**f**). The median $\theta_{crit}$ (**g**) and its relative uncertainty (**h**) across 18 ensemble members of $\theta_{crit}$ (Methods).

## Drivers of global variation in $\theta_{crit}$

To evaluate the possible mechanisms controlling the spatial variation of $\theta_{crit}$ (Fig. 2g), we excluded croplands and used random forest models (Methods) with 35 candidate factors, including soil properties, vegetation structure, plant hydraulic traits and climatic variables (Supplementary Table 2). Based on a recursive feature elimination algorithm (Methods), a subset of 11 most influential predictors were selected in the final 'best' model, which explains 74% of the global spatial variation in $\theta_{crit}$ (Fig. 3a). The aridity index, defined as the ratio of mean annual potential evapotranspiration to precipitation, was identified as the most important factor; followed by leaf area index (LAI) and the sand fraction of soil texture (Fig. 3a). This result is consistent with Bassiouni, Good[1], who evaluated the relation between critical soil water potential and aridity index based on a soil water balance model and an inverse modeling analysis. But our study rather focused on observation-based $\theta_{crit}$ and used a comprehensive set of environmental variables to identify the main drivers of global $\theta_{crit}$ variations. Partial dependence analysis further showed that $\theta_{crit}$ decreases with a higher aridity index (Fig. 3b) and sand fractions (Fig. 3d) but becomes higher at higher LAI (Fig. 3c). A lower aridity index reflects wetter climates where a higher $\theta_{crit}$ can be interpreted as an adaptation trait in view of the low risk of plants to be exposed to a water limited regime. We noted that below a leaf area index of about 2.5 m²/m², the $\theta_{crit}$ decreases; above that, further increases in LAI are less important (Fig. 3c). This suggests that low $\theta_{crit}$ in arid areas are also

related to an increasing fraction of soil exposure, highlighting the role of soil evaporation in arid areas. Thus, further evaluation and measurements of soil evaporation are needed in the future to better quantify the significance of soil evaporation in arid areas. A higher LAI being positively associated with $\theta_{crit}$ (Fig. 3c) further supports the interpretation that wetter ecosystems can sustain more leaves without compromising transpiration, given that SM rarely drops below $\theta_{crit}$ during the year. Recent studies have also shown that ecosystems with higher leaf area index have a more gradual stomatal closure in response to a SM decrease, which sustains photosynthesis in periods of low to moderate water stress[35]. On the other hand, the negative response of $\theta_{crit}$ to the sand fraction is consistent with the fact that sandy soils have lower SM wilting points[44]. Indeed, sand fraction regulates the dependence of water potential to SM and water potential is the primary driver of plant water stress[41,45]. Sandy soils have a lower soil water content for the same critical soil water potential for plant stress[46,47], which explains the negative dependence of $\theta_{crit}$ to sand fraction. Our results also show that a higher leaf nitrogen content is associated with a lower $\theta_{crit}$, consistent with the fact that plants with a higher leaf nitrogen content have a larger resistance to drought[48]. We also find that $\theta_{crit}$ shows a positive dependence on precipitation frequency but a negative dependence on shortwave radiation (Fig. 3f). More frequent precipitation events[49–51] and lower shortwave radiation help reduce water stress, and thus appear to favor an adaptation towards higher $\theta_{crit}$.

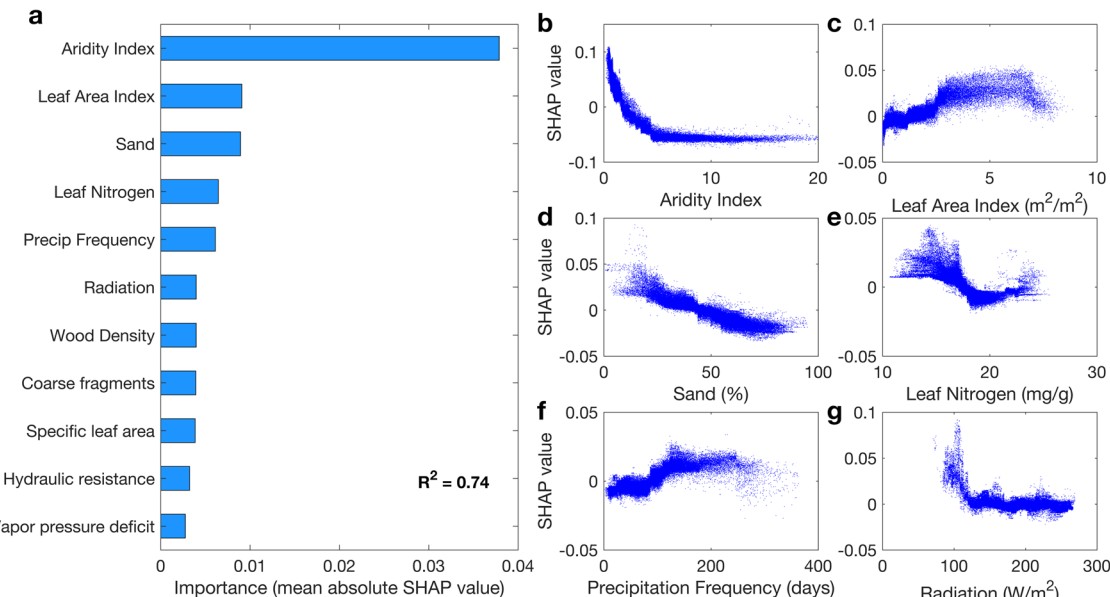

**Fig. 3 | The factors influencing global variation of critical soil moisture threshold ($\theta_{crit}$). a** The importance of climatic, biotic and edaphic variables in controlling $\theta_{crit}$. Aridity index is defined as the ratio of mean annual potential evapotranspiration to precipitation. **b**–**g** Partial dependence plots of the top six predictors. The Y-axis is SHAP value for corresponding predictor (X-axis). The partial dependence plots indicate the effects of individual variables on the response, without the influence of the other variables (Methods).

## Global distribution of the fraction of stressed days and its trend over 1979–2020

The global map of $\theta_{crit}$ (Fig. 2g) can also be used to understand how recent climate trends have affected the exposure of ecosystems to water stress. We calculated the fraction of stressed days (FSD) each year, selecting the days when SM is below $\theta_{crit}$ over each location of the globe (Methods). Combining SMAP-IB, SCA-V and SMOS-IC time series of SM during 2016–2020, we find that dryland ecosystems have a yearly FSD higher than 70% (Fig. 4a). The same analysis performed with daily surface SM and $\theta_{crit}$ from the ERA5-Land reanalysis during 2016–2020 confirms the high yearly FSD in dryland ecosystems and gives a similar spatial pattern, but a lower mean FSD (Fig. 4b). This is because the $\theta_{crit}$ estimated from ERA5-Land data is larger than that the satellite observations (Supplementary Fig. 9).

After removing the pixels with large land cover changes (>10%) to limit the impacts of land changes on trend analysis (Methods), using daily ERA5-Land SM since 1979, we found that the FSD has been increasing significantly over the last 40 years (Fig. 4c, d), implying that terrestrial ecosystems became exposed to more extensive periods of water stress. Over the past four decades, FSD increased globally, on average, by about one day per year (Fig. 4d). In addition to increased evaporative demand and atmospheric drivers[52], this increasing trend of FSD may also be attributed to increased frequency of drought and heatwaves, resulting into an overall decline of SM[53,54]. This result is in line with recent findings from Jiao, Wang[55] and Denissen, Teuling[38] based on independent data, suggesting a regime shift from energy to water limitation in relation to an overall decline of SM.

We acknowledge that $\theta_{crit}$ may change over time. Based on model outputs analysis, Hsu and Dirmeyer[56] found significant temporal variations in $\theta_{crit}$ across many locations spanning 100 years. Conversely, another study analyzed the temporal dynamics of $\theta_{crit}$ at five flux tower sites with at least 15 years of measurements and found no significant trend over time[35]. This underscores the need for future research to gain a better understanding of the temporal dynamics of $\theta_{crit}$ through longer observations. We considered here that the temporal dynamics of $\theta_{crit}$ should not hamper our trend analysis, given that even if $\theta_{crit}$ changes, its magnitude over 40 years is minimal.

## Comparison with Earth System models

Finally we diagnosed $\theta_{crit}$ by using daily EF and surface SM simulations from Earth System Models (Supplementary Table 3, Methods). We found that the models showed less spatial variability of $\theta_{crit}$ than in the observation-based map (Fig. 5a, Supplementary Figs. 11–12, Fig. 2g) and significantly underestimated $\theta_{crit}$ in wet regions (Fig. 5b, Supplementary Fig. 11), suggesting that they may underestimate the soil moisture point of inception of plant water stress in wet regions. Such a bias may lead to overly optimistic projections of the future increase of plant $CO_2$ uptake. Conversely, models significantly overestimated $\theta_{crit}$ in dry regions and failed to capture the observed very low $\theta_{crit}$ values in arid areas (Fig. 5b, Supplementary Fig. 11), which could partly explain why ESMs underestimate both gross and net $CO_2$ fluxes in dryland ecosystems[57,58]. Plants growing in arid areas have evolved many adaptation strategies to survive drought, for example by reducing leaf area index, reducing plant hydraulic and stomatal conductance, and using water stored in vegetation for transpiration[59–61]. These mechanisms have not been properly parameterized or fully integrated in models. Thus, our results can help to guide the research directions that can improve the simulation of SM stress.

We noted that the biases in $\theta_{crit}$ should not be directly equated with model accuracy in simulating water stress because the ability of models to simulate water stress not only depend on the value of $\theta_{crit}$ but also on their simulation of water uptake and transport when SM is lower than $\theta_{crit}$. For example, models reduce gas exchange at different rates when ecosystem becomes water-limited[62,63]. This leads to differences in their water and carbon simulations[64] and better $\theta_{crit}$ estimates will not resolve these differences that drive much of the water stress impacts on gross primary productivity and evapotranspiration. However, quantifying the inception of water stress – the $\theta_{crit}$, as done here, is a prerequisite for understanding the response rates of gas exchanges to SM stress. In addition, observation based models of evapotranspiration and gross primary productivity (e.g., light use efficiency models) typically assume fixed plant functional type values[65,66] to define SM stress thresholds, that are used across regions and climate. This study provides spatially explicit parameterizations of plant water stress as a function of enviromental drivers that could be

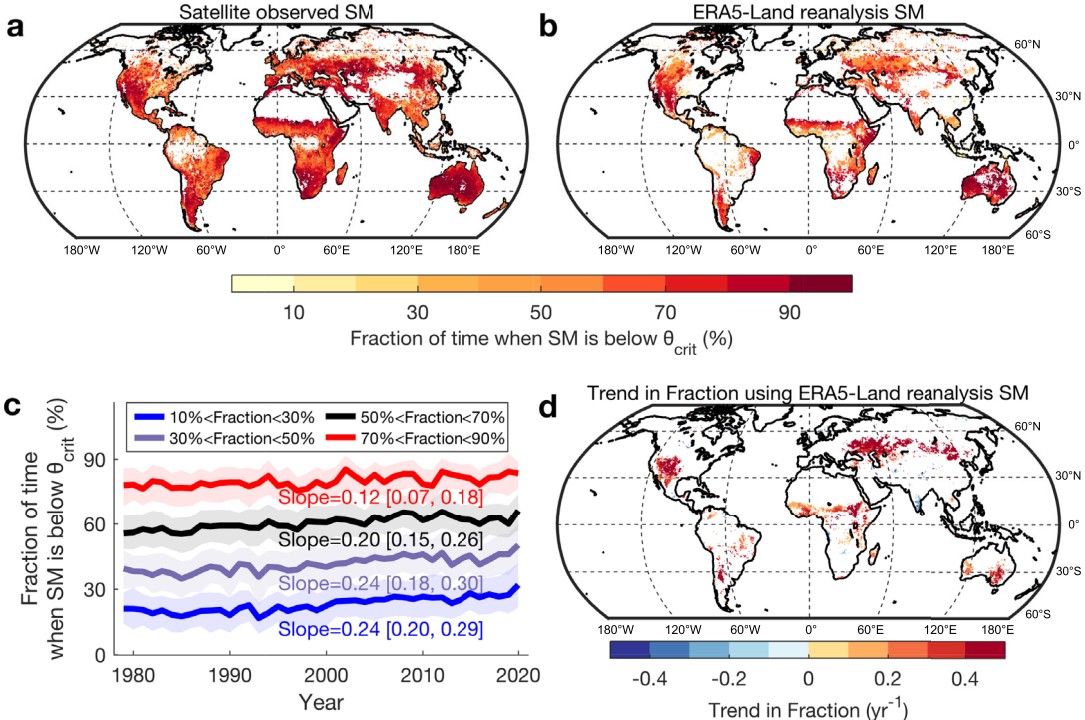

**Fig. 4 | The global distribution and trend in the fraction of time when soil moisture is below the critical soil moisture threshold ($\theta_{crit}$). a, b** The global distribution of the fraction of time when soil moisture (SM) is below $\theta_{crit}$ using satellite observed SM (median values from SMAP-IB, SCA-V and SMOS-IC) and ERA5-Land reanalysis SM during 2016–2020. **c** Annual time series of the fractions of time when SM is below $\theta_{crit}$ in regions with different fraction bins over 1979–2020. The trend (Sen's slope) and its 95% confidence interval are detected using the nonparametric trend test technique (Mann–Kendall test; $p < 0.05$). The solid line shows the median value while the shading bounds the interval of the 25th to 75th percentiles. **d** Spatial patterns of the temporal trend in the fraction of time when SM is below $\theta_{crit}$ with white indicating those areas with no significant changes (Mann–Kendall test; $p > 0.05$) or >10% land cover changes during 1982–2016[100], and colored pixels indicating areas with significant trends ($p < 0.05$).

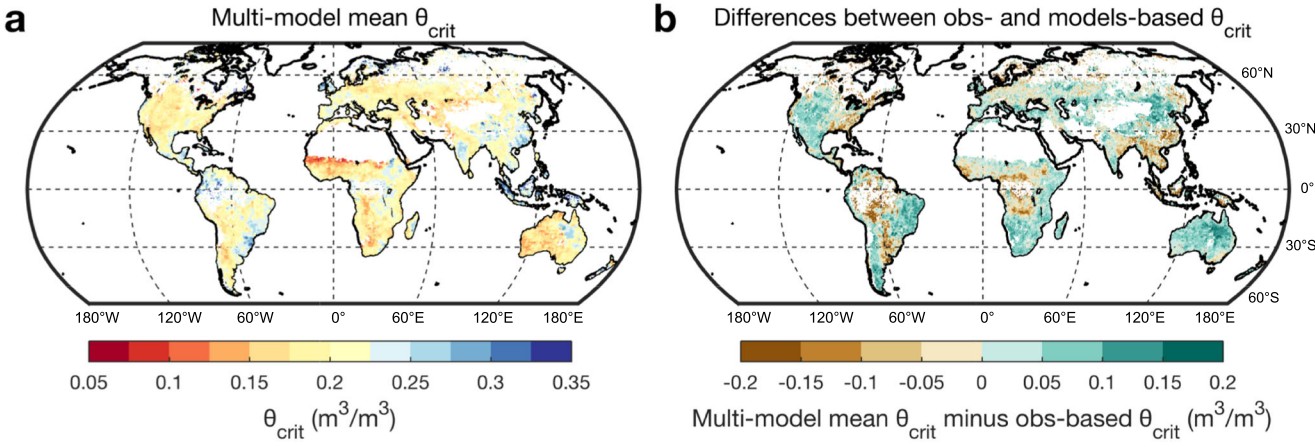

**Fig. 5 | The global distribution of critical soil moisture threshold ($\theta_{crit}$) from Earth System Models. a** The multi-model mean $\theta_{crit}$ using ten Earth System Models. **b** The differences (multi-model mean $\theta_{crit}$ minus observation-based $\theta_{crit}$) between multi-model mean $\theta_{crit}$ and observation-based $\theta_{crit}$.

incorporated in future model iterations to improve the representation of plant water stress and its spatial variations.

Vegetation regulates the terrestrial water and carbon cycles, as it controls and adapts to changing SM availability, with $\theta_{crit}$ being a key variable characterizing the coupling between soil-plant continuum and the atmosphere. Yet our ability to characterize $\theta_{crit}$ at the global scale has been limited to date. Based on the dLST–SM relationship from multiple satellite observations, this study provides the geographical distribution and assessment of the variations of $\theta_{crit}$ across the globe. We also showed the usefulness of hourly LST data from geostationary satellites to understand ecosystem water stress[67,68]. By uncovering the relationships between $\theta_{crit}$ and environmental factors, including climatic, biotic and edaphic variables, we further added mechanistic understanding of the environmental factors driving the global variation in $\theta_{crit}$. It sheds light on potential tipping points of water stress impairing ecosystem functioning, and future research will aim to use

the understanding built based on the map of $\theta_{crit}$ and its environmental drivers to improve land-surface model representation of SM constraints on water and carbon cycles. In addition, we showed that the terrestrial ecosystems experienced more frequent water-stress regimes through the past four decades, with important implications on the current land carbon sink capacity. Although the relative patterns of $\theta_{crit}$ in some models are similar to those observed, the ten state-of-the-art ESMs that we tested failed to accurately reproduce the magnitude and spatial variability of $\theta_{crit}$, suggesting the uncertain projection of current and future response of carbon uptake and evapotranspiration to droughts. These ESMs need to improve the simulation of SM and related processes, conforming to measurements, to provide more reliable projections of the response of terrestrial ecosystems to climate change and feedbacks between land and atmosphere. Together, these results demonstrated the global distribution of $\theta_{crit}$ and its drivers, applications and models' performance, with important implications for understanding the inception of water stress in models and identifying tipping points of water stress that could result in widespread impairment of ecosystem functioning and loss of ecosystem services with continued climate warming.

## Methods

### Eddy covariance measurements
We used half-hourly SM, latent heat flux, sensible heat flux, and outgoing longwave radiation from the recently released ICOS (Integrated Carbon Observation System)[69], AmeriFlux[70,71] and FLUXNET2015 datasets of energy, water, and carbon fluxes and meteorological data, all of which have undergone a standardized set of quality control and gap filling[72,73]. Data were processed following a consistent and uniform processing pipeline[72]. There were 279 flux tower sites in total by combining ICOS, AmeriFlux and FLUXNET2015 datasets. We first removed 130 sites without SM or outgoing longwave radiation measurements; then dropped all wetland sites because they have a perched water table and infrequently show SM limitations. Since for some sites, there is no dry-downdetected during the peak growing season across all available years; these sites were also excluded (81 sites remaining). The evaporative fraction (EF)−SM and land surface temperature diurnal amplitude (dLST)−SM relationships in these 81 sites were evaluated site-by-site, respectively, to detect the $\theta_{crit}$ for each site (see below). There were 44 sites with the $\theta_{crit}$ estimates for both EF−SM and dLST−SM methods. We only used the surface SM observations because surface SM (0-10 cm, varying across sites) was measured at all sites.

### Derivation of dLST from eddy covariance measurements
At each flux tower site, we derived daily dLST using measured daily maximum and minimum outgoing longwave radiation. The outgoing longwave radiation (LW) is emitted by the surface and depends on radiometric surface temperature (LST), the Stefan−Boltzmann constant ($\sigma$) and emissivity ($\varepsilon$) according to the Stefan−Boltzmann law[74] (Eq. 1). Therefore, the dLST can be calculated as Eq. 2, where $LW_{max}$ and $LW_{min}$ are the daily maximum and minimum outgoing longwave radiation, respectively; $\varepsilon$ is considered as constant at the same site and same day because we are deriving dLST (not LST).

$$LW = \varepsilon \times \sigma \times LST^4 \qquad (1)$$

$$dLST = ((LW_{max}/\sigma)^{1/4} - (LW_{min}/\sigma)^{1/4})/\varepsilon^{1/4} \qquad (2)$$

### SM and dLST from satellite observations
We used three L-band passive daily surface SM (to a depth of 5 cm) products: Soil Moisture Active Passive (SMAP)-INRAE-BORDEAUX (SMAP-IB)[20], single channel vertical polarization (SCA-V, SMAP_L3_SM_P)[75] and Soil Moisture and Ocean Salinity in version IC

(SMOS-IC)[76]. Both SMAP-IB (version 1) and SCA-V (version 7, L3 products) have 36 km resolution and one to three-day revisit from 1 April 2015 to 31 December 2020. The SMAP-IB algorithm is based on the two-parameter inversion of the L-MEB model, as defined in Wigneron, Jackson[77], applied to the SMAP mono-angular dual-polarized brightness temperature[18]. SCA-V is not independent from SMAP-IB, but their retrieval algorithms, vegetation correction and surface roughness correction are different. SCA-V is adopted as the operational baseline algorithm to estimate SM from SMAP brightness temperature[78]. In the SCA-V, vegetation is accounted for by the τ−ω model as in L-MEB. However, optical depth at nadir (τNAD) is not retrieved as for SMOS-IC and SMAP-IB. Instead it is estimated from the linear relation τNAD = b × VWC between τNAD and vegetation water content (VWC)[79]. Thereby, values of the b-parameter are assumed polarization independent and will be provided from a land cover look up table, and the VWC is estimated from values of the NDVI Index. SMOS-IC is derived from the two-parameter L-MEB inversion applied to the SMOS multi-angular and dual-polarized brightness temperatures. SMOS-IC has 25 km resolution and two to four-day revisit from 1 January 2011 to 31 December 2020[76]. Based on the recent study from Li, Wigneron[20], the biases of these three SM datasets were corrected using ISMN in-situ measurements, an international cooperation to construct and maintain a global in-situ SM database[80,81]. Across all ISMN in-situ measurements, Li, Wigneron[20] found that the biases of SMAP-IB, SCA-V and SMOS-IC are 0.002, 0.008 and −0.054 m³/m³, respectively. We thus corrected the biases of these three SM datasets by subtracting the corresponding bias.

Two land surface temperature datasets from the Copernicus Global Land Operations and Moderate Resolution Imaging Spectroradiometer (MODIS) were used. The Copernicus LST (version 2) datasets are obtained from a constellation of geostationary satellite missions: Meteosat Second Generation (MSG) and Indian Ocean Data Coverage (IODC) missions, Geostationary Operational Environmental Satellite (GOES) and Himawari (and its predecessor Multi-Function Transport Satellite - MTSAT)[82,83]. The Copernicus LST provides hourly data at a spatial resolution of 5 km covering most of the globe's land surface, but there is no geostationary coverage in parts of northern and eastern Europe, Central Asia, and the Indian subcontinent as well as parts of eastern Siberia and northern North America (Supplementary Fig. 1). The second LST datasets are from the Terra (MOD11C1) and Aqua (MYD11C1) MODIS Version 6.1 Land Surface Temperature, providing four observed LST per day (10:30 AM/PM, 1:30 AM/PM) at a 0.05-degree resolution. We calculated the daily dLST as the difference between daily maximum and minimum LST using hourly LST from Copernicus or four observed LST every day from MODIS Terra and Aqua. The bilinear interpolation algorithm was applied to resample all data into the grid resolution of 0.25 degree.

### Soil moisture dry-down identification
Dry-downs following rainfall are episodes with no rain for several consecutive days during which SM shows a short term 'pulse' rise after rain and then decays until the next rain event. At each flux tower site, a dry-down is retained for our analysis when SM decreases consecutively for at least 10 days after rainfall following previous studies[30,31,84–86]. The results were similar after requiring the soil dry-down to be at least 9 or 11 days. To ensure the reliability of latent heat flux measurements (high signal-to-noise ratio), we focused on the soil dry-downs during the peak growing season for all available site-years, defined as three-month periods with the maximum mean gross primary productivity across the available years. For satellite observations, soil drydowns were defined as at least 5 (for SMAP-IB and SCA-V, one to three-day revisit) or 4 (for SMOS-IC, two to four-day revisit) consecutive overpasses (over ≥ 10 days) of decreasing SM. The full year data of satellite observations were used and results were found to be similar when only growing season data were used.

 7

Soil dry-down periods provided the unique and consistent opportunity for us to detect the $\theta_{crit}$ because the transition from energy limitation to soil water limitation is likely to happen during the dry-down period. However, less dry-downs are available in wet regions than in dry regions. As a results, some wet regions including a small number of dry-downs were masked as it was not possible to detect the $\theta_{crit}$ in such situations. Note that, if some wet regions have experienced more frequent droughts recently, they will thus have more dry-downs and have been included in our analysis. This approach partly reduces the spatial coverage of our global map of $\theta_{crit}$, but highlights that some wet regions should be further investigated in the future with longer satellite observations.

### $\theta_{crit}$ estimation using EF–SM and dLST–SM methods

While other factors limit evapotranspiration besides SM and the linear dependency is a simple approximation, many previous studies have showed that the EF–SM and dLST–SM framework provides a good first-order representation of regimes of land–atmosphere coupling, both in models and observations (e.g., Seneviratne, Corti[5], Seneviratne, Lüthi[87], Koster, Dirmeyer[88], Koster, Suarez[89], Teuling, Seneviratne[90], Feldman, Short Gianotti[14]). Here we provided a global analysis based on this first-order theoretical and empirically verified framework. We calculated the daily EF as the ratio of observed latent heat flux to the sum of latent and sensible heat fluxes. Then, we characterized the EF–SM and dLST–SM relationship at each site, respectively, using all available soil dry-downs, from a regression between these two variables with a linear-plus-plateau model:

$$EF \text{ or } dLST = \begin{cases} a + b(SM - \theta_{crit}) \; if \; SM < \theta_{crit} \\ a \; if \; SM \geq \theta_{crit} \end{cases} \quad (3)$$

where $a$ is the maximum (or minimum) value of EF (or dLST) in the absence of SM stress (energy-limited stage), $b$ represents the slope of the linear phase (water-limited stage) between EF (or dLST) and SM, and $\theta_{crit}$ is the critical SM threshold. $\theta_{crit}$ represents the breakpoint until which EF (or dLST) increases (or decreases) linearly as a function of SM. The $\theta_{crit}$ and its standard error were simultaneously estimated by least squares fit with the R software package 'segmented'[91] for each site. An example to estimate the $\theta_{crit}$ using EF–SM and dLST–SM methods is shown in Fig. 1a, b. Following Feldman, Short Gianotti[14], we considered three models: SM varying only within a water-limited regime (linear model) or energy-limited regime (linear model), and SM varying within a transitional regime (linear-plus-plateau model). According to the lowest Akaike Information Criterion[92], we selected the "best" model pixel by pixel, and the $\theta_{crit}$ is detected when the linear-plus-plateau model is selected. Some pixels or sites did not have a defined $\theta_{crit}$ value if there were either no dry-downs or if SM varied only within a water- or energy-limited regime, thus rendering the breakpoint analysis of dLST–SM or EF–SM impossible. Based on the EF–SM and dLST–SM relationships, there were 44 sites (Supplementary Table 1) with the $\theta_{crit}$ estimates for both EF–SM and dLST–SM methods. The Pearson correlation and its associated statistical test were used to compare the $\theta_{crit}$ values from the dLST–SM method with that of EF–SM method. Increasing dLST is a direct observable signature of shifts in the surface energy partitioning regimes[25,26]. An increased diurnal temperature range, for a given amount of net radiation, is directly linked to a decrease in EF and thus increased soil moisture stress[3]. dLST is positively associated with sensible heating but negatively associated with EF and SM[27–29]. Evaporative regimes and $\theta_{crit}$ estimating have been characterized previously with observed dLST–SM relationships across some regions, such as Africa[14] and site level[28], showing that the dLST–SM relationship is an effective method to estimate $\theta_{crit}$. Here we applied this method to the global scale using multiple satellite observations.

For global satellite observations, we quantified $\theta_{crit}$ for each pixel based on dLST–SM method using all dry-downs from 1 April 2015 to 31 December 2020. There were 18 maps of $\theta_{crit}$ in total by considering all possible combinations of three SM datasets (SMAP-IB, SCA-V and SMOS-IC) and two dLST datasets (Copernicus and MODIS) and the uncertainty of $\theta_{crit}$ estimates ($\theta_{crit}$ and $\theta_{crit} \pm$ standard error, from the linear-plus-plateau model), resulting from different data sources and estimating variants. The median $\theta_{crit}$ and its relative uncertainty at each pixel were calculated across 18 ensemble members. The relative uncertainty was defined as the ratio of standard error to the median value of these ensemble members. The $\theta_{crit}$ estimated from satellite ensembles was compared with $\theta_{crit}$ estimated from flux tower sites using the dLST–SM method or the EF–SM method, respectively. For each site, we extracted and calculated the median $\theta_{crit}$ values within a $3 \times 3$ pixel window around the site from satellites-derived $\theta_{crit}$ map. The Pearson correlation and its associated statistical test were used to compare $\theta_{crit}$ based on satellite observations and flux towers across 26 sites. For the remaining 18 sites, the satellite data could not be used to derive $\theta_{crit}$ because there were either no dry-downs, or SM varied only within a water- or energy-limited regime, or the number of samples were too low, thus rendering the breakpoint analysis of dLST–SM unreliable. Note that both the measurement time periods and frequency of flux tower sites differ from those of satellites. The results show that satellite-derived $\theta_{crit}$ is significantly correlated to that estimated independently from eddy covariance measurements (Supplementary Fig. 5). The correlations between satellite-derived $\theta_{crit}$ and tower-derived $\theta_{crit}$ using both the dLST–SM method and the EF–SM method are strongly significant ($p < 0.01$), but their correlation coefficients are not very high ($r = 0.57$ for the dLST–SM method and $r = 0.55$ for the EF–SM method), which may be due to several factors. First, the footprint size ranges from a few meters to dozens of meters for the flux tower measurements but reaches 25 kilometers for satellite observations (0.25 degree). This mismatch is expected to lead to difference of $\theta_{crit}$ values estimated from flux towers and satellites. Second, the soil depths of measured SM from flux towers and the quality of SM measurements varied among different sites, and the depths from flux towers are also different from those of satellite observations. This could also contribute to the differences found between $\theta_{crit}$ values. Third, daily data from both flux towers and satellites were used, and high variability and measurement errors affect the data at this short time scale. Moreover, there are 48 measurements per day for flux towers but only a few revisits per week for satellite SM. These factors could introduce biases when comparing their $\theta_{crit}$ values. We noted that the $\theta_{crit}$ estimated from satellites is a bit higher than that of flux towers in the low $\theta_{crit}$ range (Supplementary Fig. 5), which may be attributed to higher SM values from satellite data compared to measurements from flux towers in arid regions because of different sampling depths between flux tower measurements and satellite observations.

While satellite-based $\theta_{crit}$ used surface SM, both Dong, Akbar[33] and Fu, Ciais[35] revealed that these thresholds also provide information deeper into the subsurface, and proved that surface and rootzone SM are often similarly skillful for identifying evapotranspiration regime changes based on in-situ observations. Feldman, Short Gianotti[34] recently also reported that remotely sensed surface SM can capture deep water dynamics relevant to plant water uptake so that L-band satellite SM data used here are relevant to vegetation rootzones. To further test whether the $\theta_{crit}$ obtained from surface SM also provide information deeper into the subsurface at global scale, we used dLST and SM data with different soil layers from ERA5-Land and compared the $\theta_{crit}$ values derived from ERA5-Land SM layer 1 (0–7 cm depth) with the layers 2 (7–28 cm) or 3 (28–100 cm).

### $\theta_{crit}$ among different biomes and the impacts of farm management on $\theta_{crit}$ in croplands

To compare $\theta_{crit}$ values among different biomes, the International Geosphere–Biosphere Program (IGBP) classification from MCD12C1

and Köppen climate classification map were used (Supplementary Fig. 7). We also used the aridity classification by the United Nations Environment Program[93] (Supplementary Table 2), and the global landmass is classified into five categories, namely, (i) hyperarid, (ii) arid, (iii) semi-arid, (iv) dry sub-humid, and (v) humid (Supplementary Fig. 8). We performed a more detailed analysis of the $\theta_{crit}$ differences between cropland types, based on the expectation that $\theta_{crit}$ should be affected by the choice of cultivars and by management practices such as irrigation. The geographic distribution of main staple crops was from Monfreda, Ramankutty[94]. The Global Map of Irrigation Areas (Version 5) was downloaded from the website of The Food and Agriculture Organization[95]. This map showed the amount of area equipped for irrigation in percentage of the total area on a raster. We also tested the hypothesis that the areas of recent cropland expansion over drier marginal lands should be associated with a decrease of $\theta_{crit}$, as more crop species adapted to dry environments would be selected. For cropland expansion, we used the map of percent of cropland net gain per pixel during 2003–2019 from Potapov, Turubanova[37]. Differences in $\theta_{crit}$ between groups (different biomes or climate types or farm managements in croplands) were analyzed using the Kruskal–Wallis test, a nonparametric test of difference[96]. A $p < 0.05$ was used to identify significant differences between groups.

## Drivers of global variation in $\theta_{crit}$

A random forest analysis was used to identify the factors (soil property, vegetation structure, plant hydraulic traits and climate – 35 factors in total) that contribute the most to the geographic variation in $\theta_{crit}$. These variables were chosen due to their relevance to soil and vegetation dynamics based on field studies, sites observations and their availability at the global scale. $\theta_{crit}$ is a composite attribute, reflecting soil and vegetation attributes. We investigated the importance of 35 factors (Supplementary Table 2) that reflect the dual role of soils and vegetation in determining $\theta_{crit}$[9]. Predictor variables with low predictive power were removed from the random forest models to avoid overfitting. Following Green, Ballantyne[97], we first ran a random forest model with all predictor variables included, and the predictor variables were ranked according to their permutation importance. The model was then rerun with the least important variable removed from the model, a process called recursive feature elimination (RFE)[98]. Importance values were then recalculated and stored, and this process was repeated until the three most important predictor variables remained. From here, the $R$-squared value was tabulated based on the out-of-bag observations (~one-third of the observations), and then, the model was rerun with the next most important variable added back in (based on the importance rankings stored during RFE). The $R$-squared value of this model based on the out-of-bag observations was then retabulated, and should the $R$-squared value increase by at least 0.005, the predictor variable remained in the model (otherwise, it was removed) and the next most important variable was then added back into the model and was rerun with a new $R$-squared value tabulated. This process was repeated until all predictor variables had been added back into the model, and the variable combination with the highest out-of-bag $R$-squared value was selected for the final model. Additionally, for each model, the number of variables used at each node split (between 2 and the number of predictor variables, with a final selection of 4) and the number of trees used in the model (between 50 and 5000, with a final selection of 1500) were optimized to maximize out-of-bag $R$-square value. In this way, the best quality model could be developed by only including the most informative inputs. The final set of predictors included the following 11 predictor variables: sand fraction and volumetric fraction of coarse fragments to describe soil properties; leaf area index, leaf nitrogen concentration, woody density and specific leaf area to describe vegetation structures; plant hydraulic resistance to describe plant hydraulic traits; aridity index (defined as the ratio of

mean annual potential evapotranspiration to precipitation[99]), mean annual precipitation frequency, shortwave radiation and vapor pressure deficit to describe climatic factors.

The selected random forest model was used to calculate Shapley values (SHAP), and thus analyze the sensitivity of the output to the input variables, and improve upon feature importance. Shapley values, based on game theory, assess every combination of predictors to determine each predictor's impact. For example, focusing on the aridity index feature, the approach tests the accuracy of every combination of features not including aridity index and then tests how adding aridity index improves the accuracy on each combination. Thus, the results from the partial SHAP dependency analysis can be used to determine the effects of individual variables on the response, without the influence of other variables[97].

## Calculating the fraction of stressed days and its trend

To explore how many days in a year that ecosystem are water-limited, we calculated the fraction of stressed days (FSD), defined as the ratio of the number of days with SM < $\theta_{crit}$ to the total observed daily SM in a year for each pixel. The fraction of time when SM is below $\theta_{crit}$ was computed for each year during 2016–2020 using SMAP-IB, SCA-V and SMOS-IC, respectively. Then the median value across these three SM datasets was calculated. The same analysis was also performed for the daily ERA5-Land reanalysis SM dataset because it has longer time series (1979–2020). Following satellite datasets analysis, the $\theta_{crit}$ for ERA5-Land was estimated using ERA5-Land SM and dLST. We first compared the FSD from ERA5-Land reanalysis SM during 2016–2020 with that of satellite observations. Then we calculated the FSD from ERA5-Land reanalysis SM for each year from 1979 to 2020. The overall trends (Sen's slope) of the FSD in regions with mean fractions of times spent below $\theta_{crit}$ within 10% to 30%, 30% to 50%, 50% to 70% and 70% to 90%, were detected, respectively, using the nonparametric trend test technique (Mann–Kendall test). To avoid the impacts of extreme values, we did not include the regions with mean fractions below 10% or above 90%. Additionally, we also used the Mann–Kendall test to evaluate trend (Sen's slope) in the fraction of time when SM is below $\theta_{crit}$ at each pixel and map its trend globally. A $p < 0.05$ was used to identify statistically significant trends. We noted that there are some data gaps in $\theta_{crit}$ derived from ERA5-Land datasets (Supplementary Fig. 9a) because of the failure to fit a breakpoint model, suggesting that there are some inconsistencies in SM or dLST data between ERA5-Land and satellites. The large spread of the scatters between ERA5-Land derived and satellite-derived $\theta_{crit}$ (Supplementary Fig. 9b) partly indicates such a discrepancy. We thus compared the daily ERA5-Land SM and dLST with those from satellites for a day in 2020, and found that the primary biases between ERA5-Land and satellites lie in SM data, rather than dLST data (Supplementary Fig. 10).

To remove the impacts of land cover changes on the trend analysis, we masked the pixels with >10% land cover changes during 1982–2016 according to the global land changes data from Song, Hansen[100] based on daily satellite observations acquired by the Advanced Very High Resolution Radiometer. Song, Hansen[100] quantified the global land changes during the period 1982–2016 and developed an annual vegetation continuous fields (VCF, representing the land surface as a fractional combination of vegetation functional types) product consisting of tree canopy cover, short vegetation cover and bare ground cover and characterized land change over the past 35 years (0.05-degree resolution). The bilinear interpolation algorithm was applied to resample this data into the grid resolution of 0.25 degree. We also considered that the temporal dynamics of $\theta_{crit}$ should not hamper the trend analysis because Fu, Ciais[35] have analyzed the temporal dynamics of $\theta_{crit}$ at five flux tower sites with at least 15 years of measurements, and found no significant trend with time in $\theta_{crit}$.

## CMIP6 ESM simulations

Ten ESMs (ACCESS-ESM1-5, BCC-ESM1, Can-ESM5, CMCC-CM2, INM-CM5, IPSL-CM6A, MIROC6, MPI-ESM1-2-HR, MRI-ESM2 and NorESM2-MM) in CMIP6 provided daily surface SM, latent and sensible heat fluxes outputs (Supplementary Table 3). Daily SM and calculated EF (from latent and sensible heat fluxes) from historical runs (2009–2014) were used for each model. Following the observational analysis, the same analysis was carried out for the ten CMIP6 models. For each model, we first selected all soil dry-downs from the full-year dataset of model outputs, defined as at least 10 consecutive days of decreasing SM, then quantified $\theta_{crit}$ pixel-by-pixel by means of the EF−SM relationship. Multi-model mean $\theta_{crit}$ was calculated for each pixel by averaging the $\theta_{crit}$ across these ten models. To evaluate the $\theta_{crit}$ performance in ESMs, we also calculated the difference between multi-model mean $\theta_{crit}$ and observation-based $\theta_{crit}$. We noted that different models led to different simulated SM values[101], and this inherent divergence of simulated SM distribution could also contribute to the differences between observation-based $\theta_{crit}$ and models-based $\theta_{crit}$ values. But we found that all models consistently showed less spatial variability of $\theta_{crit}$ than in the observation-based map, suggesting that our result did not depend on the inherent divergence of simulated SM distribution.

## Data availability

The global critical soil moisture thresholds of plant water stress are available at https://zenodo.org/records/11183719. The eddy covariance measurements are downloaded from the ICOS (https://doi.org/10.18160/2G60-ZHAK), AmeriFlux (https://ameriflux.lbl.gov/) and FLUXNET2015 datasets (https://fluxnet.fluxdata.org/data/fluxnet2015-dataset/). SMAP-IB and SMOS-IC SM are obtained from https://ib.remote-sensing.inrae.fr/. SCA-V are available on National Snow and Ice Data Center (https://smap.jpl.nasa.gov/data/). Copernicus LST are downloaded from https://land.copernicus.eu. MODIS LST are from https://lpdaac.usgs.gov/. The Global Map of Irrigation Areas (Version 5) was downloaded from the website of The Food and Agriculture Organization (https://www.fao.org/aquastat/en/geospatial-information/global-maps-irrigated-areas/latest-version). ERA5-Land reanalysis data are from https://cds.climate.copernicus.eu/. The CMIP6 data are downloaded from https://esgf-data.dkrz.de/search/cmip6-dkrz/.

## Code availability

The primary code used to generate the results is publicly available at https://zenodo.org/records/11183719.

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

## Acknowledgements

This work was financially supported by the IGSNRR (E3V30050), National Natural Science Foundation of China (31988102), CNES (5100019800) and the ANR CLAND Convergence Institute. I.C.P. acknowledges support by European Research Council funding under the European Union's Horizon 2020 research and innovation program (787203 REALM). P.G. and I.C.P. acknowledge support by the LEMONTREE (Land Ecosystem Models based On New Theory, observation and Experiments) project, funded through the generosity of Eric and Wendy Schmidt by recommendation of the Schmidt Futures program. A.F.F. was supported by an appointment to the NASA Postdoctoral Program at the NASA Goddard Space Flight Center, administered by Oak Ridge Associated Universities under contract with NASA. W.K.S. was supported by the NASA Carbon Cycle and Ecosystems Program under grant 80NSSC23K0109. D.M. was supported by the INRAE metaprogrammes CLIMAE and XRISQUES. A.R.K. was supported by the College of Agricultural Sciences at Penn State University via USA Hatch Appropriations under Project PEN04710 and Accession 1020049. We would like to thank the ICOS Infrastructure for support in collecting and curating the eddy covariance data. This work used global eddy covariance data acquired and shared by the FLUXNET community, including these networks: AmeriFlux, AfriFlux, AsiaFlux, CarboAfrica, CarboEuropeIP, CarboItaly, CarboMont, China-Flux, Fluxnet-Canada, GreenGrass, ICOS, KoFlux, LBA, NECC, OzFlux-TERN, TCOS-Siberia and USCCC. The ERA-Interim reanalysis data are provided by ECMWF and processed by LSCE. The FLUXNET eddy covariance data processing and harmonization were carried out by the European Fluxes Database Cluster, AmeriFlux Management Project and Fluxdata project of FLUXNET, with the support of CDIAC and ICOS Ecosystem Thematic Center and the OzFlux, ChinaFlux and AsiaFlux offices.

## Author contributions

Z.F. and P.C. designed the study. Z.F. performed the analysis. Z.F. and P.C. wrote the paper with the inputs from all co-authors. J.P.W., P.G., A.F.F., D.M., N.V., A.R.K., D.S.G., I.C.P., P.C.S., D.Y., L.Y.L., H.L.M., X.J.L., Y.Y.H., K.L.Y., P.Z., X.L., Z.C.Z., J.H.L. and W.K.S. provided methodological suggestions and contributed to the interpretation of the results.

## Competing interests

The authors declare no competing interests.

## Additional information

¹Key Laboratory of Ecosystem Network Observation and Modeling, Institute of Geographic Sciences and Natural Resources Research, Chinese Academy of Sciences, Beijing 100101, China. ²Laboratoire des Sciences du Climat et de l'Environnement, LSCE/IPSL, CEA-CNRS-UVSQ, Université Paris-Saclay, Gif-sur-Yvette 91191, France. ³ISPA, INRAE, Université de Bordeaux, Bordeaux Sciences Agro, F-33140 Villenave d'Ornon, France. ⁴Department of Earth and Environmental Engineering, Columbia University, New York, NY 10027, USA. ⁵NASA Goddard Space Flight Center, Biospheric Sciences Laboratory, Greenbelt, MD 20771, USA. ⁶Earth System Science Interdisciplinary Center, University of Maryland, College Park, MD, USA. ⁷Unit Applied Mathematics and Computer Science (UMR MIA-PS) INRAE AgroParisTech Université Paris-Saclay, Palaiseau 91120, France. ⁸Department of Plant Science, The Pennsylvania State University, 116 Agricultural Science and Industries Building, University Park, PA 16802, USA. ⁹Department of Biological Systems Engineering, University of Wisconsin—Madison, Madison, USA. ¹⁰Georgina Mace Centre for the Living Planet, Department of Life Sciences, Imperial College London, Silwood Park Campus, Buckhurst Road, Ascot SL5 7PY, UK. ¹¹Ministry of Education Key Laboratory for Earth System Modeling, Department of Earth System Science, Tsinghua University, Beijing 100084, China. ¹²Earth and Planetary Sciences, Weizmann Institute of Science, Rehovot 7610001, Israel. ¹³INRAE, Avignon Universit´e, UMR 1114 EMMAH, UMT CAPTE, F-84000 Avignon, France. ¹⁴Department of Geography, The University of Hong Kong, Hong Kong, SAR, China. ¹⁵Research Institute of Agriculture and Life Sciences, Seoul National University, Seoul, South Korea. ¹⁶Peking University Shenzhen Graduate School, Peking University, Shenzhen 518055 Guangdong, China. ¹⁷School of Natural Resources and the Environment, University of Arizona, Tucson, AZ, USA. ✉e-mail: fuzheng@igsnrr.ac.cn

