## [Peer Review File · Nature Communications]

Global critical soil moisture thresholds of plant water stressEditorial Note: Parts of this Peer Review File have been redacted as indicated to remove third-party material where no permission to publish could be obtained.

REVIEWER COMMENTS

Reviewer #1 (Remarks to the Author):

The authors present an analysis of the 'critical soil moisture threshold' (θ_{crit}) based on remotely sensed and model data and validated with tower data. The θ_{crit} value is where soil moisture becomes limiting and evapotranspiration (ET) fluxes begin to decrease. Understanding the value of θ_{crit} can help in creating and parameterizing models of vegetation stress and its effects on ET. There have been numerous recent studies of θ_{crit} , including a similar study the authors have previously conducted with global sets of tower-based observation (see ref 6 in particular). The main addition here appears to be doing a similar thing with satellite data as well as comparing with θ_{crit} derived from model outputs and evaluating trends in θ_{crit} through time.

Overall, the study is well written and the subject matter is of interest to a wide audience. I suggest a few interconnected issues which should be addressed.

1) Given the number of prior studies in this subject area (e.g. refs 1, 6, 7, 16, and others cited) the authors do not do a sufficient job of placing the finding here within the context of these earlier works. These prior studies also examined θ_{crit} , or similar quantities, and so how is what is learned here different, similar, and building upon this prior body of work. We're similar patterns observed with respect to aridity, LAI, veg type, etc.? This is lacking in the manuscript.

2) In a similar vein, little is present about what hydrologic theory would suggest these values are. For instance plant wilting points and other soil moisture thresholds are often defined with respect to the soil water potential. Wilting points are typically specified as being the same wilting point across textures, with the volumetric water content different from this based on soil retention curves. While I see that the authors have soil textures, they should be able to translate their observations into water potentials which would represent more physical properties driving the movement of water.

3) Similarly ecological theory about plant stress in various environments has a long history. What does this body of literature say we should expect in more (or less) arid environments or different ecosystems. Which types of ecosystems are expected to shut down quicker or later? This has been approached in a variety of ways including LSM model parameterization, remote sensing of isohydrity, and other studies on vegetation stress thresholds, and so on. What new information about these trends/patterns/relationships by the study the authors present.

All the above points are interconnected and represent a lack of placement of this study's results within the context of prior work. Given that these θ_{crit} values are not directly to be used in models (if they are then the authors need to compare against what values models are currently using) a better effort needs to be made to demonstrate the value of the work done here. This can be achieved by properly building off of prior understanding.

Reviewer #2 (Remarks to the Author):

This paper presents a global assessment of the critical soil moisture thresholds using satellite-based soil moisture and land surface temperature (LST) estimates, unveiling global patterns of plant hydraulic strategies adapted to atmospheric dryness and soil water availability. A global map of critical soil moisture thresholds was derived by analyzing the relationship between diurnal LST range and soil moisture during dry-down periods (days after rainfall). The approach is validated with in-situ data from global flux towers (44 sites). The study showed that Earth System Models underestimate the spatial variability of the critical soil moisture threshold, indicating potential model deficiencies. Lastly, the paper revealed an increasing trend in the fraction of days with soil moisture below the critical threshold during the past four decades based on ERA5-Land reanalysis data.

This study identifies the critical soil water threshold, its global patterns, and the environmental factors,

which provide important insights into ecosystem water-stress regimes and soil moisture constraints on global water cycles. The paper is well-written, and the results are clearly and effectively presented. However, I have a few questions and comments regarding the potential confounding factors for critical soil moisture threshold determination and the analysis of ERA5-Land data, that I hope the authors could clarify or consider.

First, the analysis could be enhanced by a thorough evaluation of the confounding variables affecting the dLST-SM and EF-SM relationships, thereby, the temporal variation of critical soil moisture threshold. In this study, a single soil moisture threshold was derived for each pixel/site, yet it is possible that such a threshold present temporal variation linked to atmosphere and vegetation changes. As suggested by Feldman et al. (2019, ref 16), VPD consistently mediates the threshold inferred from the dLST-SM relationship in African grasslands. Therefore, it would be beneficial to include confounding variables in the statistical analysis and understand whether and how they may affect spatial patterns of critical soil moisture threshold.

Secondly, the extensive data gaps in the soil moisture threshold derived from ERA5-Land data presented, as seen in Fig. 4 and Fig. 9, require a more detailed explanation. The extent of missing data, which is much higher than satellite-based analysis, is quite surprising since ERA5-Land is a reanalysis product. I'm interested in understanding the main cause of data gaps: lack of dry-down periods, limited range of SM, or missing data. If the main cause is the failure of the breakpoint model, it may suggest that there are major inconsistencies in SM or LST data between ERA5-Land and satellite estimates. The large spread of the scatters between ERA5-Land derived and satellite-derived θ_{crit} in Supplementary Fig. 9b partly indicates such a discrepancy. An explanation and discussion of these inconsistencies would be helpful to strengthen the analysis.

Another minor suggestion is to include a simple plot of θ_{crit} between satellite-derived v.s. ESM-model estimates, perhaps summarized by biome or climate types. This could provide a more straightforward illustration of how ESMs underestimate the threshold in wet regions and overestimate the threshold in water-limited regions than the maps shown in Fig. 5.

Minor comments:

Main text

L164-165: This is an interesting hypothesis. A more detailed elaboration on how recently expanded cropland is associated with lower critical soil moisture than established ones would be helpful.

Fig. 4: More context on the ERA5_Land analysis would be helpful for clarity. Did the ERA5-Land analysis use the θ_{crit} derived from ERA5-Land? Which relationship was used, dLST-SM or ET-SM?

L263: While the ESMs' θ_{crit} estimates show a much smaller spatial variability, the relative patterns are quite similar to that of the satellite-derived values, at least in some models.

Method

L295: Does daily minimum LW occur during nighttime?

L302: To avoid confusion, consider using the official product name for "SCA-V" (e.g., SMAP_L3_XX). I was initially confused that "SCA-V" is soil moisture retrieved from another satellite, before reading the method second in detail.

L316: It would be helpful to provide the temporal availability of the SMOS-IC dataset.

L389: Could you please double-check the reference here? After a brief read of Reference 7, I didn't find relevant results on dLST-SM relationship, but maybe I missed it. In any case, a brief summary of previous work on dLST-SM for θ_{crit} estimating would be helpful.

L411: I am curious if there are any potential explanations about the consistently higher estimates from satellite versus flux tower in the low θ_{crit} range.

L417: Could you please elaborate on the third point about how the high temporal variations could lead to uncertainties in the estimates?

L467: It would be helpful to have the final selected hyperparameters as a reference for other and

future studies.

Reviewer #3 (Remarks to the Author):

Fu et al. introduce a novel approach for estimating the critical soil moisture threshold and the fraction of water-limited days, deviating from methods previously limited to reanalysis and climate models. The global map generated in this study is of significant value, establishing a foundational reference for future comparisons. Given my own focus on soil moisture regime and critical values estimation, I find the figures in this study particularly exciting. The writing is logical, and the methods are clearly articulated. While I have only a few comments and suggestions, I recommend acceptance of the paper after minor revisions. Hsin Hsu (Wish not to remain anonymous)

1.The paper shows that the Aridity Index is a primary determinant of spatial variation in θ_{crit} . This also suggests that annual variability in Aridity Index can lead to annual variability in θ_{crit} . However, the annual FSD is calculated based on climatological θ_{crit} rather than year-specific values. This approach potentially compromises the fidelity of the FSD trend estimation. Although the authors reference a study (Line 507) indicating limited changes in θ_{crit} , another research (Hsu and Dirmeyer, 2023) suggests significant changes in θ_{crit} over many locations in one century. The FSD examination of this study (40 year- reanalysis-global) is between the two mentioned extreme cases (15 year-observation-few sites vs. 100 year with +1%CO₂ per year in climate model). Given that the period used in all of these studies are within similar increasing rate of CO₂, the evidence of 15-year observation analysis might not effectively support authors' statement. The authors could consider a decadal θ_{crit} estimation from ERA5-land as supplementary information and/or moving the argument of temporal variation of θ_{crit} from method to main text with a more comprehensive discussion.

Hsu and Dirmeyer, 2023. Uncertainty in Projected Critical Soil Moisture Values in CMIP6 Affects the Interpretation of a More Moisture-Limited World. Doi: 10.1029/2023EF003511

2.Maybe I miss this somewhere: The paper lacks clarity regarding the sampling method for drydown used to estimate θ_{crit} for climate models. It is unclear whether the author selects identical months as observations or the model's peak growing seasons. Discussing the implications of this choice is essential, as differences in background climate (and the aridity index) between observations and model outputs may lead to variations in estimated θ_{crit} .

3.Regarding the comment #3 on divergence of hydroclimate among data products. how does author think the inherent divergence of simulated soil moisture distribution (Koster et al. 2009) affects the interpretation here. Does this extends beyond model-specific vegetation/soil dynamics as discussed in lines 231-236?

Koster et al. 2009: On the Nature of Soil Moisture in Land Surface Models. Doi: 10.1175/2009JCLI2832.1

4.While the method using dLST appears straightforward, the main text could benefit from a physical description of the connection between using Evaporative Fraction (EF) and dLST, explaining why this approach is effective (Line 84-85). This addition would enhance the audience's understanding of the methodology.

Line 77. typo: atmosphere

**Global critical soil moisture thresholds of plant water stress
NCOMMS-23-53911-T**

Response to Reviewers

We greatly appreciate the opportunity to revise our manuscript, and would like to thank all reviewers for their valuable expertise and constructive comments. We have carefully revised the manuscript following the reviewers' suggestions. Consequently, our manuscript has been considerably improved. Please see below our point-by-point responses in blue text following reviewer comments. The line numbers referred to are for the clean version of the revised manuscript (non-track-change version).

Reviewer #1 (Remarks to the Author):

1.1 The authors present an analysis of the 'critical soil moisture threshold' (θ_{crit}) based on remotely sensed and model data and validated with tower data. The θ_{crit} value is where soil moisture becomes limiting and evapotranspiration (ET) fluxes begin to decrease. Understanding the value of θ_{crit} can help in creating and parameterizing models of vegetation stress and its effects on ET.

There have been numerous recent studies of θ_{crit} , including a similar study the authors have previously conducted with global sets of tower-based observation (see ref 6 in particular). The main addition here appears to be doing a similar thing with satellite data as well as comparing with θ_{crit} derived from model outputs and evaluating trends in θ_{crit} through time.

Overall, the study is well written and the subject matter is of interest to a wide audience. I suggest a few interconnected issues which should be addressed.

Thank you very much for supporting this paper and recognizing the significance of our study for a wide audience. We have carefully revised the manuscript following your suggestions. Consequently, our manuscript has been considerably improved. Please see below our detailed responses.

1.2 1) Given the number of prior studies in this subject area (e.g. refs 1, 6, 7, 16, and others cited) the authors do not do a sufficient job of placing the finding here within the context of these earlier works. These prior studies also examined θ_{crit} , or similar quantities, and so how is what is learned here different, similar, and building upon this prior body of work. We're similar patterns observed with respect to aridity, LAI, veg type, etc.? This is lacking in the manuscript.

Thank you very much for this constructive comment. We agree that prior studies investigated this subject area. Bassiouni, Good¹ examined the critical soil water potential threshold based on a soil water balance model and an inverse modeling framework. Several previous observational studies also reported θ_{crit} at flux tower sites² or regional level³. Although these previous works contributed to improve our knowledge on θ_{crit} at sites level and highlighted the importance of scaling observed θ_{crit} globally, they did not provide global maps of θ_{crit} , due to a lack of global

high-frequency observations of EF, leaving a gap in our understanding of θ_{crit} at the global scale. Please also note that current land surface models are using soil moisture rather than soil water potential.

Going beyond previous studies, we expended an approach initially developed over Africa to generate the first observation-based global map of θ_{crit} by combining systematic satellite observations and in-situ data from flux towers. The approach to map θ_{crit} is based on the diurnal amplitude of the land surface temperature and daily soil moisture measurements during periods of consecutive days without rain, using in-situ flux tower networks and satellite observations from different sensors, including geostationary satellites to measure the diurnal amplitude of land surface temperature. This observation-based map of θ_{crit} allowed us to evaluate the possible mechanisms controlling the global variation of θ_{crit} , and to diagnose simulated θ_{crit} from current Earth System Models. Our study thus evaluated the global distribution of observation-based θ_{crit} and its drivers. We also show that models differ greatly from observations, and that models have spatially too smooth gradients of θ_{crit} , underestimating θ_{crit} in drylands and overestimating it in wet biomes. The novelties of this study include:

1. A global comparison of two approaches to estimate θ_{crit} values: Evaporative Fraction decreases during soil moisture dry-downs, which only applicable at in situ flux tower sites, and land surface temperature diurnal amplitude increases during soil moisture dry-downs, applicable globally from satellite observations;
2. The first observation-based global map of θ_{crit} , derived by combining satellite observations and in-situ data from flux towers.
3. Exploration of the drivers of global θ_{crit} variations using a very comprehensive set of drivers and explainable machine learning;
4. Uncovering the distribution of the fraction of stressed days when soil moisture stays below θ_{crit} and its trend over 1979-2020;
5. Applied the framework to CMIP6 models and diagnosed the θ_{crit} from current Earth System Models simulations;
6. Brought together multiple lines of evidence to better understand θ_{crit} and its implications.

Following your suggestions, we have added more discussion of previous works and highlighted our findings and novelties in the revised manuscript.

Line 79-82:

“Some model-based analyses have used the concept of critical soil water potential^{1, 4}, but current land surface models are using soil moisture rather than soil water potential, and global observation-based analyses of critical thresholds are still missing.”

Line 202-205:

“This result is consistent with Bassiouni, Good¹, who evaluated the relation between critical soil water potential and aridity index based on a soil water balance model and an inverse modeling analysis. But our study rather focused on observation-based θ_{crit} and used a comprehensive set of environmental variables to identify the main drivers of global θ_{crit} variations.”

Line 215-217:

“Recent studies have also shown that ecosystems with higher leaf area index have a more gradual stomatal closure in response to a SM decrease, which sustains photosynthesis in periods of low to moderate water stress².”

Line 92-93:

“Evaporative regimes have been characterized with observed dLST–SM relationships across Africa, but not yet globally³, leaving a gap in our understanding of θ_{crit} across the globe.”

Line 150-155:

“We further use SM data with different soil layers from ERA5-Land (Methods) and compare the θ_{crit} values derived from ERA5-Land SM layer 1 (0-7 cm depth), layer 2 (7-28 cm) and layer 3 (28-100 cm). We found that surface θ_{crit} is highly correlated with θ_{crit} derived from deep soil layers (Supplementary Fig. 6), showing that θ_{crit} obtained from surface SM can provide information deeper into the subsurface, consistent with the results of flux tower observations reported by both Dong, Akbar⁵ and Fu, Ciais².”

Line 179-192:

“The spatial distribution of θ_{crit} in this study aligns with previous findings in ecological theory regarding plant stress across various environments^{1, 3, 4, 5, 9, 10, 11}. Land surface models often have a lower θ_{crit} model parameter in arid biomes^{8, 9, 11}. The map of ecosystem-scale isohydricity from remotely sensed observations showed that the anisohydric behavior is more common in arid ecosystems¹⁰. By quantifying the soil water potential threshold, Bassiouni, Good¹ showed that water uptake strategies in arid locations are generally more drought resistant. Note that soil water potential is rarely measured in situ, and land surface models are using soil moisture rather than soil water potential. Different vegetation water stress in arid and humid ecosystems have also been recognized in many other studies, based on the ecosystem limitation index⁹, the Land Surface Water Index^{12, 13}, and SM anomalies¹¹. However, these indicators are not direct measures of water stress. The θ_{crit} values quantified in our study reflect the long-term adaptation of ecosystems to aridity regimes. θ_{crit} is simple to define and is a direct measure of water stress, but θ_{crit} remains not observed and our study allows to compare it across biomes. θ_{crit} can also be used to quantify the time spent below θ_{crit} and understand how recent climate trends have affected the exposure of ecosystems to water stress.”

Line 284-304:

“Based on the dLST–SM relationship from multiple satellite observations, this study provides the geographical distribution and assessment of the variations of θ_{crit} across the globe. We also showed the usefulness of hourly LST data from geostationary satellites to understand ecosystem water stress^{6, 7}. By uncovering the relationships between θ_{crit} and environmental factors, including climatic, biotic and edaphic variables, we further added mechanistic understanding of the environmental factors driving the global variation in θ_{crit} . It sheds light on potential tipping points of water stress impairing ecosystem functioning, and future research will aim to use the new understanding built based on the map of θ_{crit} and its environmental drivers to improve land-surface model representation of SM constraints on water and carbon cycles. In addition, we showed that the terrestrial ecosystems experienced more frequent water-stress regimes through the past four decades, with important implications on the current land carbon sink capacity. The ten state-of-the-art ESMs that we tested failed to accurately reproduce the magnitude of θ_{crit} ,

suggesting the uncertain projection of current and future response of carbon uptake and evapotranspiration to droughts. These ESMS need to improve the simulation of SM and related processes, conforming to measurements, to provide more reliable projections of the response of terrestrial ecosystems to climate change and feedbacks between land and atmosphere. Together, these results demonstrated the global distribution of θ_{crit} and its drivers, applications and models' performance, with important implications for understanding the inception of water stress in models and identifying tipping points of water stress that could result in widespread impairment of ecosystem functioning and loss of ecosystem services with continued climate warming.”

1.3 2) In a similar vein, little is present about what hydrologic theory would suggest these values are. For instance plant wilting points and other soil moisture thresholds are often defined with respect to the soil water potential. Wilting points are typically specified as being the same wilting point across textures, with the volumetric water content different from this based on soil retention curves. While I see that the authors have soil textures, they should be able to translate their observations into water potentials which would represent more physical properties driving the movement of water.

Thank you for this critical comment. We fully agree that soil water potential represents better the physical properties in driving the movement of water than soil moisture. Following your suggestions, we tried to converted our map of θ_{crit} into a map of critical soil water potential (ψ_{crit}) using soil pedotransfer functions, which showed the map of ψ_{crit} is not realistic, with more negative ψ_{crit} values in the driest regions and less negative ones in wet ecosystems (Figure R1). The bias is likely due to the uncertainty in soil pedotransfer functions and parameters. It is well known that the parameter distributions in pedotransfer functions are poorly constrained and prevent confident transformation of soil moisture to soil water potential. For example, even relatively small variations in a single parameter of the van Genuchten model cause soil water potential to vary by an order of magnitude over a wide range of soil moisture⁸. Please look at the ref⁸ for more details about the large uncertainty in converting soil moisture to soil water potential using soil pedotransfer functions and parameters. Given the large uncertainties in soil pedotransfer functions and parameters, we choose not to add the map of critical soil water potential into the manuscript.

Figure R1. Estimated soil matrix potential threshold by converting the θ_{crit} based on soil pedotransfer function and parameters.

1.4 3) Similarly ecological theory about plant stress in various environments has a long history. What does this body of literature say we should expect in more (or less) arid environments or different ecosystems. Which types of ecosystems are expected to shut down quicker or later? This has been approached in a variety of ways including LSM model parameterization, remote sensing of isohydricity, and other studies on vegetation stress thresholds, and so on. What new information about these trends/patterns/relationships by the study the authors present.

Thank you for this valuable suggestion. The spatial distribution of θ_{crit} in this study is in agreement with previous studies in ecological theory about plant stress in various environments^{1, 3, 4, 5, 9, 10, 11}. Land surface models often have a lower θ_{crit} model parameter in arid biomes^{8, 9, 11}. The map of ecosystem-scale isohydricity derived from remotely sensed observations showed that the anisohydric behavior is more common in arid ecosystems¹⁰. Note that this plant trait is not a direct indicator of water stress. By quantifying the soil water potential threshold, Bassiouni, Good¹ showed that water uptake strategies in arid locations are generally more drought resistant. But soil water potential is rarely measured in situ, and land surface models are using soil moisture rather than soil water potential. Different vegetation water stress in arid and humid ecosystems have also been recognized in many other studies, based on the ecosystem limitation index⁹, the Land Surface Water Index^{12, 13}, and SM anomalies¹¹. However, these indicators are not direct measures of water stress.

Quantifying the inception of water stress – the θ_{crit} , as done in our study, is a prerequisite for understanding the response rates of gas exchanges to SM stress. θ_{crit} is simple to define and is a direct measure of water stress (Evaporative Fraction drops), but θ_{crit} remains not observed and our study allows to compare it across biomes. The θ_{crit} values quantified in our study reflect the long-term adaptation of ecosystems to aridity regimes. Ecosystems with low θ_{crit} often have greater adaptive capacity to deal with water stress because low θ_{crit} reflects a resistance to soil dryness. In addition, explainable machine learning models (random forest) were applied in our study to gain insights on the climatic, biotic and edaphic factors controlling the spatial variations of θ_{crit} . Based on our map of θ_{crit} , we can also calculate the fraction of stressed days (FSD, Fig. 4 in the manuscript) to explore how many days in a year that ecosystem are water-limited.

Observation based models of evapotranspiration and gross primary productivity (e.g., light use efficiency models) typically assume fixed plant functional type values^{14, 15} to define SM stress thresholds, that are used across regions and climate. Our study provides spatially explicit parameterizations of plant water stress as a function of environmental drivers that could be incorporated in future model iterations to improve the representation of plant water stress and its spatial variations.

In the revised manuscript, we now added this discussion as suggested.

Line 179-192:

“The spatial distribution of θ_{crit} in this study aligns with previous findings in ecological theory regarding plant stress across various environments^{1, 3, 4, 5, 9, 10, 11}. Land surface models often have a lower θ_{crit} model parameter in arid biomes^{8, 9, 11}. The map of ecosystem-scale isohydricity from remotely sensed observations showed that the anisohydric behavior is more common in arid

ecosystems¹⁰. By quantifying the soil water potential threshold, Bassiouni, Good¹ showed that water uptake strategies in arid locations are generally more drought resistant. Note that soil water potential is rarely measured in situ, and land surface models are using soil moisture rather than soil water potential. Different vegetation water stress in arid and humid ecosystems have also been recognized in many other studies, based on the ecosystem limitation index⁹, the Land Surface Water Index^{12, 13}, and SM anomalies¹¹. However, these indicators are not direct measures of water stress. The θ_{crit} values quantified in our study reflect the long-term adaptation of ecosystems to aridity regimes. θ_{crit} is simple to define and is a direct measure of water stress, but θ_{crit} remains not observed and our study allows to compare it across biomes. θ_{crit} can also be used to quantify the time spent below θ_{crit} and understand how recent climate trends have affected the exposure of ecosystems to water stress.”

1.5 All the above points are interconnected and represent a lack of placement of this studies results within the context of prior work. Given that these theta_crit values are not directly to be used in models (if they are then the authors need to compare against what values models are currently using) a better effort needs to be made to demonstrate the value of the work done here. This can be achieved by properly building off of prior understanding.

Thank you very much for these constructive comments. In this study, we are trying our best and making the most of current knowledges to generate the first observation-based global map of θ_{crit} by combining systematic satellite observations and in-situ data from flux towers. We have followed your suggestions and carefully revised the manuscript as above. As a consequence, our manuscript has been considerably improved. Please see our detailed responses above. We hope that our responses and revision of the manuscript are satisfactory to you.

Reviewer #2 (Remarks to the Author):

2.1 This paper presents a global assessment of the critical soil moisture thresholds using satellite-based soil moisture and land surface temperature (LST) estimates, unveiling global patterns of plant hydraulic strategies adapted to atmospheric dryness and soil water availability. A global map of critical soil moisture thresholds was derived by analyzing the relationship between diurnal LST range and soil moisture during dry-down periods (days after rainfall). The approach is validated with in-situ data from global flux towers (44 sites). The study showed that Earth System Models underestimate the spatial variability of the critical soil moisture threshold, indicating potential model deficiencies. Lastly, the paper revealed an increasing trend in the fraction of days with soil moisture below the critical threshold during the past four decades based on ERA5-Land reanalysis data.

This study identifies the critical soil water threshold, its global patterns, and the environmental factors, which provide important insights into ecosystem water-stress regimes and soil moisture constraints on global water cycles. The paper is well-written, and the results are clearly and effectively presented. However, I have a few questions and comments regarding the potential confounding factors for critical soil moisture threshold determination and the analysis of ERA5-Land data, that I hope the authors could clarify or consider.

We greatly appreciate the reviewer's positive comments and support of the manuscript. We have thoroughly revised the manuscript following your suggestions. Consequently, our manuscript has been considerably improved. Please see below our detailed responses.

2.2 First, the analysis could be enhanced by a thorough evaluation of the confounding variables affecting the dLST-SM and EF-SM relationships, thereby, the temporal variation of critical soil moisture threshold. In this study, a single soil moisture threshold was derived for each pixel/site, yet it is possible that such a threshold present temporal variation linked to atmosphere and vegetation changes. As suggested by Feldman et al. (2019, ref 16), VPD consistently mediates the threshold inferred from the dLST-SM relationship in African grasslands. Therefore, it would be beneficial to include confounding variables in the statistical analysis and understand whether and how they may affect spatial patterns of critical soil moisture threshold.

Thank you for this critical comment. While other factors limit evapotranspiration besides SM and the linear dependency is a simple approximation, many previous studies have showed that the EF-SM framework provides a good first-order representation of regimes of land-atmosphere coupling (Fig. 5 from Seneviratne, Corti ¹⁶), both in models and observations (e.g., Seneviratne, Corti ¹⁶, Seneviratne, Lüthi ¹⁷, Koster, Dirmeyer ¹⁸, Koster, Suarez ¹⁹, Teuling, Seneviratne ²⁰, Feldman, Short Gianotti ³). Here we provided a global analysis based on this first-order theoretical and empirically verified framework.

[Redacted]

Fig. 5 from Seneviratne, Corti ¹⁶. Definition of soil moisture regimes and corresponding evapotranspiration regimes. This paper has been cited over 4000 times by the field since 2010.

We fully agree that it is possible that the critical soil moisture threshold present temporal variation linked to atmospheric and vegetation changes. But as Feldman et al. 2019 concluded that, the actual impacts of these confounding factors (e.g. VPD, net radiation) on θ_{crit} are minor, and that soil moisture remains to be the dominant factor in partitioning surface energy in the transition from non-water limited to water limited regimes. It is also important to note that Feldman et al. 2019's analysis on confounding factors quantified how short-term changes of confounding factors influenced dLST–SM relationships. As stated in Feldman et al. 2019, caveats exist in the analysis of confounding factors, including high covariation between regressors, assumption of linear interactions, and inability to account for auto-regressive terms due to irreconcilable time gaps between drydowns. Given these uncertainties, we choose to not conduct such an analysis. Instead, we used random forest models and uncovered the relationships between θ_{crit} and environmental factors, including climatic, biotic and edaphic factors, to evaluate the possible drivers in controlling the spatial variation of θ_{crit} . We believe this is much more important than considering the minor effects of confounding factors on θ_{crit} based on short-term temporal variations.

In the revised manuscript, we now added this discussion according to your suggestion.

Line 392-397:

“While other factors limit evapotranspiration besides SM and the linear dependency is a simple approximation, many previous studies have showed that the EF–SM and dLST–SM framework provides a good first-order representation of regimes of land–atmosphere coupling, both in models and observations (e.g., Seneviratne, Corti ¹⁶, Seneviratne, Lüthi ¹⁷, Koster, Dirmeyer ¹⁸, Koster, Suarez ¹⁹, Teuling, Seneviratne ²⁰, Feldman, Short Gianotti ³). Here we provided a global analysis based on this first-order theoretical and empirically verified framework.”

2.3 Secondly, the extensive data gaps in the soil moisture threshold derived from ERA5-Land data presented, as seen in Fig. 4 and Fig. 9, require a more detailed explanation. The extent of missing data, which is much higher than satellite-based analysis, is quite surprising since ERA5-Land is a reanalysis product. I'm interested in understanding the main cause of data gaps: lack of dry-down periods, limited range of SM, or missing data. If the main cause is the failure of the breakpoint model, it may suggest that there are major inconsistencies in SM or LST data between ERA5-Land and satellite estimates. The large spread of the scatters between ERA5-Land derived

and satellite-derived θ_{crit} in Supplementary Fig. 9b partly indicates such a discrepancy. An explanation and discussion of these inconsistencies would be helpful to strengthen the analysis.

Thank you for this constructive comment. We compared the daily ERA5-Land SM and dLST with those from satellites for a day in 2020 (Supplementary Fig. 10). We found that the primary biases between ERA5-Land and satellites lie in SM data, rather than dLST data, emphasizing the importance of global soil moisture measurements. Following your suggestions, we have added the explanation and discussion in the revised manuscript.

Supplementary Fig. 10. Comparison of daily ERA5-Land SM and dLST with those from satellites for a day in 2020.

Line 537-543:

“We noted that there are some data gaps in θ_{crit} derived from ERA5-Land datasets (Supplementary Fig. 9a) because of the failure to fit a breakpoint model, suggesting that there are some inconsistencies in SM or dLST data between ERA5-Land and satellites. The large spread of the scatters between ERA5-Land derived and satellite-derived θ_{crit} (Supplementary Fig. 9b) partly indicates such a discrepancy. We thus compared the daily ERA5-Land SM and dLST with those from satellites for a day in 2020, and found that the primary biases between ERA5-Land and satellites lie in SM data, rather than dLST data (Supplementary Fig. 10).”

2.4 Another minor suggestion is to include a simple plot of θ_{crit} between satellite-derived v.s. ESM-model estimates, perhaps summarized by biome or climate types. This could provide a more straightforward illustration of how ESMs underestimate the threshold in wet regions and overestimate the threshold in water-limited regions than the maps shown in Fig. 5.

Following your interesting suggestion, we now show a new plot comparing the distributions of θ_{crit} derived from satellites vs ESMs, summarized by climate.

Supplementary Fig. 11. Comparison between multi-model mean θ_{crit} and observation-based θ_{crit} grouped by climate types based on the aridity classification.

Line 257-264:

“We found that the models showed less spatial variability of θ_{crit} than in the observation-based map (Fig. 5a, Supplementary Figs. 11-12, Fig. 2g) and significantly underestimated θ_{crit} in wet regions (Fig. 5b, Supplementary Fig. 11), suggesting that they may underestimate the soil moisture point of inception of plant water stress in wet regions. Such a bias may lead to overly optimistic projections of the future increase of plant CO₂ uptake. Conversely, models significantly overestimated θ_{crit} in dry regions and failed to capture the observed very low θ_{crit} values in arid areas (Fig. 5b, Supplementary Fig. 11), which could partly explain why ESMs underestimate both gross and net CO₂ fluxes in dryland ecosystems^{21, 22}.”

Minor comments:

Main text

2.5 L164-165: This is an interesting hypothesis. A more detailed elaboration on how recently expanded cropland is associated with lower critical soil moisture than established ones would be helpful.

Thank you for this comment. Following your suggestion, we have added more explanations in the revised manuscript.

Line 476-478:

“We also tested the hypothesis that the areas of recent cropland expansion over drier marginal lands should be associated with a decrease of θ_{crit} , as more crop species adapted to dry environments would be selected.”

2.6 Fig. 4: More context on the ERA5_Land analysis would be helpful for clarity. Did the

ERA5-Land analysis use the θ_{crit} derived from ERA5-Land? Which relationship was used, dLST-SM or ET-SM?

Yes, the ERA5-Land analysis used the θ_{crit} derived from ERA5-Land SM and dLST. We now stated it clearly in the revised manuscript as suggested.

Line 528-529:

“Following satellite datasets analysis, the θ_{crit} for ERA5-Land was estimated using ERA5-Land SM and dLST.”

Line 712-713:

“The global distribution of θ_{crit} using ERA5-Land surface SM and dLST.”

2.7 L263: While the ESMs’ θ_{crit} estimates show a much smaller spatial variability, the relative patterns are quite similar to that of the satellite-derived values, at least in some models.

Thank you for this suggestion. We have modified the sentence to read “Although the relative patterns of θ_{crit} in some models are similar to those observed, the ten state-of-the-art ESMs that we tested failed to accurately reproduce the magnitude and spatial variability of θ_{crit} , suggesting the uncertain projection of current and future response of carbon uptake and evapotranspiration to droughts.”

Method

2.8 L295: Does daily minimum LW occur during nighttime?

The daily minimum LW occurs either in the morning (e.g., 5 am) or at night (e.g., 19 pm), and this timing varies across different sites and days.

2.9 L302: To avoid confusion, consider using the official product name for “SCA-V” (e.g., SMAP_L3_XX). I was initially confused that “SCA-V” is soil moisture retrieved from another satellite, before reading the method second in detail.

Thank you for this suggestion. We have added the official product name for “SCA-V” (SMAP_L3_SM_P) as suggested.

Line 334-336:

“We used three L-band passive daily surface SM (to a depth of 5 cm) products: Soil Moisture Active Passive (SMAP)-INRAE-BORDEAUX (SMAP-IB)²³, single channel vertical polarization (SCA-V, SMAP_L3_SM_P)²⁴ and Soil Moisture and Ocean Salinity in version IC (SMOS-IC)²⁵.”

2.10 L316: It would be helpful to provide the temporal availability of the SMOS-IC dataset.

Changed as the reviewer suggested.

Line 348-349:

“SMOS-IC has 25 km resolution and two to four-day revisit from 1 January 2011 to 31

December 2020²⁵.”

2.11 L389: Could you please double-check the reference here? After a brief read of Reference 7, I didn't find relevant results on dLST-SM relationship, but maybe I missed it. In any case, a brief summary of previous work on dLST-SM for θ_{crit} estimating would be helpful.

Thank you for your careful reading. We have deleted the reference 7 here and added a brief summary of previous work per your suggestions.

Line 420-423:

“Evaporative regimes and θ_{crit} estimating have been characterized previously with observed dLST–SM relationships across some regions, such as Africa³ and site level²⁶, showing that the dLST–SM relationship is an effective method to estimate θ_{crit} . Here we applied this method to the global scale using multiple satellite observations.”

Line 88-92:

“Specifically, the land-surface temperature diurnal amplitude (dLST) starts to increase below θ_{crit} when ecosystems plunge into the water-limited regime^{27, 26, 28, 29}. An increased dLST, for a given amount of net radiation, is directly linked to a decrease in EF and thus increased SM stress³⁰. dLST is positively associated with sensible heating but negatively associated with EF and SM^{26, 28, 29}.”

2.12 L411: I am curious if there are any potential explanations about the consistently higher estimates from satellite versus flux tower in the low θ_{crit} range.

Thank you for this comment. We have added the possible explanations in the revised manuscript per your suggestion.

Line 452-455:

“We noted that the θ_{crit} estimated from satellites is a bit higher than that of flux towers in the low θ_{crit} range (Supplementary Fig. 5), which may be attributed to higher SM values from satellite data compared to measurements from flux towers in arid regions because of different sampling depths between flux tower measurements and satellite observations.”

2.13 L417: Could you please elaborate on the third point about how the high temporal variations could lead to uncertainties in the estimates?

Thank you for this suggestion. We have elaborated on the third point in the revised manuscript as suggested.

Line 449-452:

“Third, daily data from both flux towers and satellites were used, and high variability and measurement errors affect the data at this short time scale. Moreover, there are 48 measurements per day for flux towers but only a few revisits per week for satellite SM. These factors could introduce biases when comparing their θ_{crit} values.”

2.14 L467: It would be helpful to have the final selected hyperparameters as a reference for other and future studies.

Changed as the reviewer suggested.

Line 504-507:

“Additionally, for each model, the number of variables used at each node split (between 2 and the number of predictor variables, with a final selection of 4) and the number of trees used in the model (between 50 and 5000, with a final selection of 1500) were optimized to maximize out-of-bag R -square value.”

Reviewer #3 (Remarks to the Author):

3.1 Fu et al. introduce a novel approach for estimating the critical soil moisture threshold and the fraction of water-limited days, deviating from methods previously limited to reanalysis and climate models. The global map generated in this study is of significant value, establishing a foundational reference for future comparisons. Given my own focus on soil moisture regime and critical values estimation, I find the figures in this study particularly exciting. The writing is logical, and the methods are clearly articulated. While I have only a few comments and suggestions, I recommend acceptance of the paper after minor revisions. Hsin Hsu (Wish not to remain anonymous)

We appreciate Dr. Hsu's positive comments and very helpful suggestions. We have thoroughly revised the manuscript following your suggestions. Please see below our detailed responses.

3.2 1. The paper shows that the Aridity Index is a primary determinant of spatial variation in θ_{crit} . This also suggests that annual variability in Aridity Index can lead to annual variability in θ_{crit} . However, the annual FSD is calculated based on climatological θ_{crit} rather than year-specific values. This approach potentially compromises the fidelity of the FSD trend estimation. Although the authors reference a study (Line 507) indicating limited changes in θ_{crit} , another research (Hsu and Dirmeyer, 2023) suggests significant changes in θ_{crit} over many locations in one century. The FSD examination of this study (40 year- reanalysis-global) is between the two mentioned extreme cases (15 year-observation-few sites vs. 100 year with +1%CO₂ per year in climate model). Given that the period used in all of these studies are within similar increasing rate of CO₂, the evidence of 15-year observation analysis might not effectively support authors' statement. The authors could consider a decadal θ_{crit} estimation from ERA5-land as supplementary information and/or moving the argument of temporal variation of θ_{crit} from method to main text with a more comprehensive discussion.

Hsu and Dirmeyer, 2023. Uncertainty in Projected Critical Soil Moisture Values in CMIP6 Affects the Interpretation of a More Moisture-Limited World. Doi: 10.1029/2023EF003511

Thank you for this constructive comment. Following your suggestion, we have moved the argument of temporal variation of θ_{crit} from method to main text with a more comprehensive discussion by referring to Hsu and Dirmeyer³¹.

Line 247-253:

“We acknowledge that θ_{crit} may change over time. Based on model outputs analysis, Hsu and Dirmeyer³¹ found significant temporal variations in θ_{crit} across many locations spanning 100 years. Conversely, another study analyzed the temporal dynamics of θ_{crit} at five flux tower sites with at least 15 years of measurements and found no significant trend over time². This underscores the need for future research to gain a better understanding of the temporal dynamics of θ_{crit} through longer observations. We considered here that the temporal dynamics of θ_{crit} should not hamper our trend analysis, given that even if θ_{crit} changes, its magnitude over 40 years is minimal.”

3.3 2. Maybe I miss this somewhere: The paper lacks clarity regarding the sampling method for

drydown used to estimate θ_{crit} for climate models. It is unclear whether the author selects identical months as observations or the model's peak growing seasons. Discussing the implications of this choice is essential, as differences in background climate (and the aridity index) between observations and model outputs may lead to variations in estimated θ_{crit} .

Thank you for this suggestion. The full year data of satellite observations were used and results were found to be similar when only growing season data were used. Following satellite datasets analysis, the full year data of model outputs were also used. This is now clearly stated in the revised manuscript.

Line 560-564:

“Following the observational analysis, the same analysis was carried out for the ten CMIP6 models. For each model, we first selected all soil dry-downs from the full-year dataset of model outputs, defined as at least 10 consecutive days of decreasing SM, then quantified θ_{crit} pixel-by-pixel by means of the EF–SM relationship.”

3.4 3. Regarding the comment #3 on divergence of hydroclimate among data products. how does author think the inherent divergence of simulated soil moisture distribution (Koster et al. 2009) affects the interpretation here. Does this extends beyond model-specific vegetation/soil dynamics as discussed in lines 231-236?

Koster et al. 2009: On the Nature of Soil Moisture in Land Surface Models. Doi: 10.1175/2009JCLI2832.1

Thank you for this critical comment. We agree that different models led to different simulated soil moisture values, which could contribute to the differences between observation-based θ_{crit} and models-based θ_{crit} values. But we found that all models consistently showed less spatial variability of θ_{crit} than in the observation-based map (Supplementary Fig. 12), suggesting that our result did not depend on the inherent divergence of simulated soil moisture distribution. Please also note that we only used the surface soil moisture for all models so that the effects of the inherent divergence of simulated soil moisture from different models are muted.

Following your suggestion, we have discussed the effects of inherent divergence of simulated soil moisture distribution from different models by referring to Koster, Guo ³².

Line 566-570:

“We noted that different models led to different simulated SM values³², and this inherent divergence of simulated SM distribution could also contribute to the differences between observation-based θ_{crit} and models-based θ_{crit} values. But we found that all models consistently showed less spatial variability of θ_{crit} than in the observation-based map, suggesting that our result did not depend on the inherent divergence of simulated SM distribution.”

3.5 4. While the method using dLST appears straightforward, the main text could benefit from a physical description of the connection between using Evaporative Fraction (EF) and dLST, explaining why this approach is effective (Line 84-85). This addition would enhance the audience's understanding of the methodology.

Thank you for your valuable suggestion. We have now incorporated additional details on the connection between using EF and dLST in the revised manuscript, as per your recommendation.

Line 88-92:

“Specifically, the land-surface temperature diurnal amplitude (dLST) starts to increase below θ_{crit} when ecosystems plunge into the water-limited regime^{27, 26, 28, 29}. An increased dLST, for a given amount of net radiation, is directly linked to a decrease in EF and thus increased SM stress³⁰. dLST is positively associated with sensible heating but negatively associated with EF and SM^{26, 28, 29}”

3.6 Line 77. typo: atmosphere

Changed as suggested.

References mentioned in the responses

1. Bassiouni M, Good SP, Still CJ, Higgins CW. Plant water uptake thresholds inferred from satellite soil moisture. *Geophysical Research Letters* **47**, e2020GL087077 (2020).
2. Fu Z, *et al.* Critical soil moisture thresholds of plant water stress in terrestrial ecosystems. *Science Advances* **8**, eabq7827 (2022).
3. Feldman AF, Short Gianotti DJ, Trigo IF, Salvucci GD, Entekhabi D. Satellite - based assessment of land surface energy partitioning – soil moisture relationships and effects of confounding variables. *Water Resources Research* **55**, 10657-10677 (2019).
4. Bassiouni M, Higgins CW, Still CJ, Good SP. Probabilistic inference of ecohydrological parameters using observations from point to satellite scales. *Hydrology and Earth System Sciences* **22**, 3229-3243 (2018).
5. Dong J, Akbar R, Short Gianotti DJ, Feldman AF, Crow WT, Entekhabi D. Can Surface Soil Moisture Information Identify Evapotranspiration Regime Transitions? *Geophysical Research Letters*, e2021GL097697 (2022).
6. Xiao J, Fisher JB, Hashimoto H, Ichii K, Parazoo NC. Emerging satellite observations for diurnal cycling of ecosystem processes. *Nat Plants* **7**, 877-887 (2021).
7. Wen J, Fisher JB, Parazoo NC, Hu L, Litvak ME, Sun Y. Resolve the Clear - Sky Continuous Diurnal Cycle of High - Resolution ECOSTRESS Evapotranspiration and Land Surface Temperature. *Water Resources Research* **58**, e2022WR032227 (2022).
8. Novick KA, *et al.* Confronting the water potential information gap. *Nat Geosci* **15**, 158-164 (2022).
9. Denissen J, *et al.* Widespread shift from ecosystem energy to water limitation with climate change. *Nature Climate Change* **12**, 677-684 (2022).
10. Konings AG, Gentine P. Global variations in ecosystem-scale isohydricity. *Global Change Biology* **23**, 891-905 (2017).
11. Li X, *et al.* Global variations in critical drought thresholds that impact vegetation. *National Science Review* **10**, nwad049 (2023).
12. Du J, *et al.* Synergistic satellite assessment of global vegetation health in relation to ENSO - induced droughts and pluvials. *Journal of Geophysical Research: Biogeosciences* **126**, e2020JG006006 (2021).
13. Chandrasekar K, Sessa Sai M, Roy P, Dwevedi R. Land Surface Water Index (LSWI) response to rainfall and NDVI using the MODIS Vegetation Index product. *Int J Remote Sens* **31**, 3987-4005 (2010).
14. Kolby Smith W, *et al.* Large divergence of satellite and Earth system model estimates of global terrestrial CO₂ fertilization. *Nature climate change* **6**, 306-310 (2016).
15. Pei Y, *et al.* Evolution of light use efficiency models: Improvement, uncertainties, and implications. *Agricultural and Forest Meteorology* **317**, 108905 (2022).
16. Seneviratne SI, *et al.* Investigating soil moisture–climate interactions in a changing climate: A review. *Earth-Science Reviews* **99**, 125-161 (2010).

17. Seneviratne SI, Lüthi D, Litschi M, Schär C. Land–atmosphere coupling and climate change in Europe. *Nature* **443**, 205 (2006).
18. Koster RD, *et al.* Regions of strong coupling between soil moisture and precipitation. *Science* **305**, 1138-1140 (2004).
19. Koster RD, *et al.* Realistic initialization of land surface states: Impacts on subseasonal forecast skill. *Journal of Hydrometeorology* **5**, 1049-1063 (2004).
20. Teuling A, Seneviratne SI, Williams C, Troch P. Observed timescales of evapotranspiration response to soil moisture. *Geophysical Research Letters* **33**, (2006).
21. MacBean N, *et al.* Dynamic global vegetation models underestimate net CO2 flux mean and inter-annual variability in dryland ecosystems. *Environmental Research Letters* **16**, 094023 (2021).
22. Wang L, *et al.* Dryland productivity under a changing climate. *Nature Climate Change*, 1-14 (2022).
23. Li X, *et al.* A new SMAP soil moisture and vegetation optical depth product (SMAP-IB): Algorithm, assessment and inter-comparison. *Remote Sensing of Environment* **271**, 112921 (2022).
24. Jackson TJ. III. Measuring surface soil moisture using passive microwave remote sensing. *Hydrological processes* **7**, 139-152 (1993).
25. Wigneron J-P, *et al.* SMOS-IC data record of soil moisture and L-VOD: Historical development, applications and perspectives. *Remote Sensing of Environment* **254**, 112238 (2021).
26. Amano E, Salvucci GD. Detection and use of three signatures of soil-limited evaporation. *Remote sensing of environment* **67**, 108-122 (1999).
27. Panwar A, Kleidon A, Renner M. Do surface and air temperatures contain similar imprints of evaporative conditions? *Geophysical Research Letters* **46**, 3802-3809 (2019).
28. Betts AK, Desjardins R, Worth D, Beckage B. Climate coupling between temperature, humidity, precipitation, and cloud cover over the Canadian Prairies. *Journal of Geophysical Research: Atmospheres* **119**, 13,305-313,326 (2014).
29. Thakur G, Schymanski SJ, Mallick K, Trebs I, Sulis M. Downwelling longwave radiation and sensible heat flux observations are critical for surface temperature and emissivity estimation from flux tower data. *Scientific reports* **12**, 1-14 (2022).
30. Gentine P, Chhang A, Rigden A, Salvucci G. Evaporation estimates using weather station data and boundary layer theory. *Geophysical Research Letters* **43**, 11,661-611,670 (2016).
31. Hsu H, Dirmeyer PA. Uncertainty in Projected Critical Soil Moisture Values in CMIP6 Affects the Interpretation of a More Moisture - Limited World. *Earth's Future* **11**, e2023EF003511 (2023).
32. Koster RD, Guo Z, Yang R, Dirmeyer PA, Mitchell K, Puma MJ. On the nature of soil moisture in land surface models. *Journal of Climate* **22**, 4322-4335 (2009).

REVIEWERS' COMMENTS

Reviewer #1 (Remarks to the Author):

The authors have addressed my comments in their revision. Nice work.

Reviewer #2 (Remarks to the Author):

Dear authors,

Thank you very much for thoroughly addressing my previous comments! All my concerns have been comprehensively resolved and the paper has been significantly improved in clarity. I just have a few minor suggestions for the authors to consider incorporating.

Line 536-537 (track-change manuscript): Thanks for the clarification. Could you please also specify the temporal range for this analysis? I am curious to know whether it covers 2016-2020 (consistent with satellite-based analysis) or the entire time period of ERA5-Land.

Supplementary Fig. 9, Fig.10: The scatter plots might benefit from using a point density plot (or adjusting the transparency of the points) to mitigate the visual bias caused by overlapping. This overlap seems to visually amplify the impact of outliers and reduce the actual significance of the correlations.

Thanks again for your efforts in revising the manuscript and addressing my comments!
Congratulations on completing an exciting paper!

Reviewer #2 (Remarks on code availability):

The authors have provided a ReadMe file that provides detailed information in running the code to reproduce the results. Yet, since the raw data is public and not included in the repository, I suggest providing a more detailed description of the raw data format and preprocessing processes (if any) to facilitate reproducing. For example, the first step, "SM dry-down identification.m" reads in "*.mat" files, it would be beneficial to briefly outline the content of these files as comment or in the ReadMe.

Reviewer #3 (Remarks to the Author):

I appreciate the authors for addressing the comments and modifying their manuscript accordingly. The revised manuscript extensively discusses the relevant literature and uncertainty of their analysis, and provides great detail on their methods. I close my paper-review work and anticipate the paper's publication.

Minor comment:

The information provided in Line 310: "have undergone a standardized set of quality control and gap filling" is highly redundant to Line 320: "Data were quality controlled so that only measured..."

-Hsin Hsu

**Global critical soil moisture thresholds of plant water stress
#NCOMMS-23-53911A**

Response to Reviewers

Reviewer #1 (Remarks to the Author):

1.1 The authors have addressed my comments in their revision. Nice work.

Thank you very much for your constructive comments in the process.

Reviewer #2 (Remarks to the Author):

2.1 Thank you very much for thoroughly addressing my previous comments! All my concerns have been comprehensively resolved and the paper has been significantly improved in clarity. I just have a few minor suggestions for the authors to consider incorporating.

We greatly appreciate the reviewer's positive comments and support of the manuscript. We have thoroughly revised the manuscript following your suggestions. Please see below our detailed responses.

2.2 Line 536-537 (track-change manuscript): Thanks for the clarification. Could you please also specify the temporal range for this analysis? I am curious to know whether it covers 2016-2020 (consistent with satellite-based analysis) or the entire time period of ERA5-Land.

Thank you very much for your suggestions. As per your suggestions, we have specified the temporal range in Line 525-526: "We first compared the FSD from ERA5-Land reanalysis SM during 2016-2020 with that of satellite observations."

2.3 Supplementary Fig. 9, Fig.10: The scatter plots might benefit from using a point density plot (or adjusting the transparency of the points) to mitigate the visual bias caused by overlapping. This overlap seems to visually amplify the impact of outliers and reduce the actual significance of the correlations.

Thank you for this constructive comment. Following your suggestion, we have adjusted the transparency of the points in Supplementary Figs. 9-10.

2.4 Thanks again for your efforts in revising the manuscript and addressing my comments! Congratulations on completing an exciting paper!

Thank you for your kind words and valuable suggestions.

2.5 Reviewer #2 (Remarks on code availability):

The authors have provided a ReadMe file that provides detailed information in running the code to reproduce the results. Yet, since the raw data is public and not included in the repository, I suggest providing a more detailed description of the raw data format and preprocessing processes

(if any) to facilitate reproducing. For example, the first step, "SM dry-down identification.m" reads in "*.mat" files, it would be beneficial to briefly outline the content of these files as comment or in the ReadMe.

Thank you for this suggestion. Following your suggestion, we have added the detailed description of the raw data format and preprocessing processes in the ReadMe file. As an example, the 'SM_SCA-V_2020.mat' and 'Delta_LST_2020.mat' files have been included in the repository.

Reviewer #3 (Remarks to the Author):

3.1 I appreciate the authors for addressing the comments and modifying their manuscript accordingly. The revised manuscript extensively discusses the relevant literature and uncertainty of their analysis, and provides great detail on their methods. I close my paper-review work and anticipate the paper's publication.

Minor comment:

The information provided in Line 310: "have undergone a standardized set of quality control and gap filling" is highly redundant to Line 320: "Data were quality controlled so that only measured..."

-Hsin Hsu

Thank you for your positive comments and helpful suggestions. As per your suggestions, we have removed Line 320: "Data were quality controlled so that only measured..."